# Spin polarized Fe₁−Ti pairs for highly efficient electroreduction nitrate to ammonia

Jie Dai [1,6], Yawen Tong [2,6], Long Zhao [1,6], Zhiwei Hu [3], Chien-Te Chen [4], Chang-Yang Kuo [4,5], Guangming Zhan [1], Jiaxian Wang [1], Xingyue Zou [1], Qian Zheng [1], Wei Hou [1], Ruizhao Wang [1], Kaiyuan Wang [1], Rui Zhao [1], Xiang-Kui Gu [2]✉, Yancai Yao [1]✉ & Lizhi Zhang [1]✉

Electrochemical nitrate reduction to ammonia offers an attractive solution to environmental sustainability and clean energy production but suffers from the sluggish *NO hydrogenation with the spin−state transitions. Herein, we report that the manipulation of oxygen vacancies can contrive spin−polarized Fe₁−Ti pairs on monolithic titanium electrode that exhibits an attractive $NH_3$ yield rate of 272,000 $\mu g\,h^{-1}\,mg_{Fe}^{-1}$ and a high $NH_3$ Faradic efficiency of 95.2% at −0.4 V vs. RHE, far superior to the counterpart with spin−depressed Fe₁−Ti pairs (51000 $\mu g\,h^{-1}\,mg_{Fe}^{-1}$) and the mostly reported electrocatalysts. The unpaired spin electrons of Fe and Ti atoms can effectively interact with the key intermediates, facilitating the *NO hydrogenation. Coupling a flow−through electrolyzer with a membrane-based $NH_3$ recovery unit, the simultaneous nitrate reduction and $NH_3$ recovery was realized. This work offers a pioneering strategy for manipulating spin polarization of electrocatalysts within pair sites for nitrate wastewater treatment.

Ammonia ($NH_3$) as the critical feedstocks for artificial fertilizers production and a carbon−free energy carrier is of great significance to the modern society[1,2]. Nowadays, global $NH_3$ demand exceeds 150 million tons per year, which heavily relies on the energy−intensive Haber −Bosch process associated with the consumption of 1%–2% global energy, concurrently contributing to 1.4% of global carbon emissions[3,4]. Recently, electrochemical reduction of nitrogen ($N_2$) to $NH_3$, inspired by the natural microbial $N_2$ fixation, has attracted tremendous interest[5–7]. However, the robust N≡N (941 kJ/mol) and poor solubility in water medium of $N_2$ lead to a low $NH_3$ yield rate (less than 10 mmol $g_{cat}^{-1}\,h^{-1}$) and partial current densities (less than 1 mA cm⁻²)[8,9], hindering its widespread commercialization. Alternatively, nitrate ($NO_3^-$) can be dissociated into deoxygenated species with a much lower energy of 204 kJ/mol, making it as a more suitable nitrogen source for $NH_3$ electrosynthesis[10,11], especially regarding that $NO_3^-$ is widely distributed in industrial wastewaters and polluted ground-water, resulting in the eutrophication and the disturbance of ecosystems[12,13]. Therefore, selective reduction of $NO_3^-$ to $NH_3$ (NITRR) offers a promising route to synchronously relieve energy and environmental crisis.

Generally, NITRR initiates from the adsorption of nitrate (*NO₃) on electrodes followed by the sequential deoxygenation of $NO_3^-$ to adsorbed nitrite (*NO₂) and nitric oxide (*NO)[14,15]. Subsequently, *NO hydrogenates to the hydrogenated *NHO/*NOH species with a spin state transition[16–18]. Finally, the hydrogenated species can stepwise hydrogenate to the hydroxylamine (*NH₂OH) and the targeted product

[1]School of Environmental Science and Engineering, Shanghai Jiao Tong University, Shanghai 200240, China. [2]School of Power and Mechanical Engineering, Wuhan University, Wuhan 430072, China. [3]Max Planck Institute for Chemical Physics of Solids, Nothnitzer Strasse 40, 01187 Dresden, Germany. [4]National Synchrotron Radiation Research Center, 101 Hsin-Ann Road, Hsinchu 300092 Taiwan, China. [5]Department of Electrophysics, National Yang Ming Chiao Tung University, Hsinchu, Taiwan, China. [6]These authors contributed equally: Jie Dai, Yawen Tong, Long Zhao. ✉e-mail: xiangkuigu@whu.edu.cn; yyancai@sjtu.edu.cn; zhanglz@ccnu.edu.cn

of *NH_3[14,15,19]. It has been reported that both the adsorption of $NO_3^-$ and the subsequent hydrogenation of *NO are crucial for the rapid NITRR[10,20–24], and thus the enhancement of $NO_3^-$ adsorption and/or the acceleration of *NO hydrogenation would be effective to improve the activity of NITRR. Currently, the enhanced $NO_3^-$ adsorption has been widely reported by the surface modification[10], heteroatom doping[20], and alloying[21] of electrodes. While the acceleration of spin-transition related *NO hydrogenation via properly manipulation the spin states of electrocatalysts is seldom investigated, although this process is similar with reported oxygen-related electrocatalytic reactions such as oxygen evolution reaction (OER) and oxygen reduction reaction (ORR) processes[25,26], and generally demands tremendous energy and is kinetically sluggish, thus strongly retarding the transformation of $NO_3^-$-to-$NH_3$. Therefore, it is urgent to design advanced spin-polarized electrocatalysts to boost the spin-dependent electron transfer for superior NITRR.

Spin-polarization of metal active sites has been reported to be an effective way to accelerate the spin-state transition between reactant intermediates, because spin-polarized metal active sites are able to drive quantum spin exchange interaction and offer a spin electron transfer channel towards spin-dependent electrocatalytic reaction[27–29]. In light of this principle, paramagnetic iron (Fe) has been extensively investigated, especially for oxygen-related electrocatalytic reactions (e.g., OER and ORR), owing to its well-tunable spin degree of electronic freedom[29–32]. For instance, low spin ($t_{2g}^5 e_g^0$) and intermediate spin states ($t_{2g}^4 e_g^1$) of Fe are more favorable to penetrate the antibonding orbital of $O_2$ as compared with the high spin state of Fe ($t_{2g}^3 e_g^2$), thus facilitating the spin transition from the single-state $OH^-$/$H_2O$ to triplet-state $O_2$[31,32]. More impressively, Fe single atoms on carbon matrix were found to be effective for NITRR[9], exhibiting decent nitrate electroreduction capacity with an $NH_3$ Faradaic efficiency of ~75% and a yield rate of ~20000 $\mu g\,h^{-1}\,mg_{cat}^{-1}$. Obviously, finely manipulating the spin polarization effect of Fe-based materials can provide a solution to superior NITRR.

In this study, Fe single atoms were anchored on the inherent surface oxide layer of titanium foam, where the oxygen vacancies (OVs) of titanium oxide could trigger the synchronous spin polarization of Fe and the adjacent Ti atoms. The designed spin-polarized Fe_1-Ti pairs delivered a high $NH_3$ Faradic efficiency of 95.2% and an impressive $NH_3$ yield rate of 272000 $\mu g\,h^{-1}\,mg_{Fe}^{-1}$ at −0.4 V vs. RHE for NITRR, far superior to the counterpart monolithic electrode with spin-depressed Fe_1-Ti pairs (51000 $\mu g\,h^{-1}\,mg_{Fe}^{-1}$) and the mostly reported NITRR electrocatalysts. The spin electrons located in the 3d orbitals of Fe and Ti atoms in the spin-polarized Fe-Ti pair sites were found to dominantly contribute this excellent performance via interacting with the key intermediates, boosting the deoxygenation of $NO_3^-$ and the hydrogenation of *NO. The developed monolithic electrode with spin-polarized Fe_1-Ti pairs was also employed to construct a NITRR flow-through electrolyzer coupled with a membrane separation unit for a high-efficient nitrate conversion at an industrial-level current intensity and in-situ high-purity ammonia recovery from nitrate-containing wastewater.

## Results

### Manipulating spin polarization of Fe_1 − Ti pairs

Fe single atoms anchored on a monolithic Ti electrode with different spin polarization degrees were prepared by regulating the OVs of the inherent surface oxide layer of titanium foam, as illustrated in Fig. 1a, b. The core of the synthesis process lies in the control of the reductive/ oxidative gaseous environment during the thermal treatment (Supplementary Fig. 1, See details in Methods). Specifically, a reductive atmosphere ($H_2/Ar$) was performed to create oxygen vacancies (OVs) in the inherent oxide layer of Ti electrode to evoke the spin polarization of Fe single atoms and adjacent Ti atoms (Supplementary Fig. 2), thus producing spin-polarized Fe_1-Ti pairs (SP-Fe_1-Ti), while an oxidative atmosphere ($O_2/Air$) was employed to avoid the generation of

OVs to obtain the spin-depressed Fe_1-Ti pairs (SD-Fe_1-Ti) for comparison. During hydrogen treatment of inherent oxide layer of Ti foam, electrons were first transferred from hydrogen (H) atoms to the oxygen (O) atoms in the lattice of inherent oxide layer. Then, the lattice O leaves with the H atom to form $H_2O$, as evidenced by an obvious $H_2O$ evolution peak at about 370 °C during the temperature programmed reaction (TPR) measurement of pristine Ti foam in 5% $H_2/Ar$ (Supplementary Fig. 3), and the OVs form on the surface of TiO_x/Ti foam. Electron spin resonance (ESR) spectra was utilized to identify the electronic states trapped in OVs (Fig. 1c), which clearly showed the successful creation of OVs in the SP-Fe_1-Ti electrode, and the absence of OVs in SD-Fe_1-Ti was also confirmed by its silent ESR signals[33]. As evidenced by high-resolution transmission electron microscopy (HRTEM) images of SP-Fe_1-Ti and SD-Fe_1-Ti (Supplementary Figs. 4, 5), the distinct boundaries were observed between the interior crystalline Ti and surface amorphous titanium oxide layer (TiO_x), consistent with our previous work[34]. The aberration-corrected high-angle annular darkfield scanning transmission electron microscopy (HAADF-STEM) image demonstrated that Fe single atoms (marked by white circles) were well dispersed in the oxide layer of Ti foam for both SP-Fe_1-Ti and SD-Fe_1-Ti (Fig. 1d and Supplementary Fig. 6), and the Fe clusters or nanoparticles were not observed, consistent with the absence of metallic Fe diffraction peaks in the X-ray diffraction (XRD) pattern (Supplementary Fig. 7). The 3D surface intensity profile along the yellow arrow further confirmed the atomically dispersed Fe on the TiO_x/Ti foam (Supplementary Fig. 8). The elemental EDS mapping images revealed the uniform dispersion of Fe atoms in SP-Fe_1-Ti and SD-Fe_1-Ti (Fig. 1e and Supplementary Fig. 9).

We further utilized X-ray absorption fine structure spectrometry (XAFS) to confirm the atomic dispersion of Fe atoms on TiO_x/Ti. As evidenced in Fig. 1f, the Fourier-transformed Fe K-edge extended X-ray adsorption fine structure (FT-EXAFS) spectroscopy of SP-Fe_1-Ti manifested a dominant peak at 1.5 Å, corresponding to Fe-O coordination in the first shell[35]. The absence of Fe-Fe scattering path at 2.2 Å ruled out the metallic Fe aggregation in SP-Fe_1-Ti[35]. We also employed the wavelet-transform (WT)-EXAFS analysis to offer resolution in both radial distance and k-space and supplement the FT-EXAFS analysis. As shown in Fig. 1g, the WT-EXAFS spectra of SP-Fe_1-Ti exhibited the peak with a maximum intensity at 7.2 $Å^{-1}$ in contrast to Fe foil (Supplementary Fig. 10), which was assigned to the Fe-O scattering path. Similar locally fine structure of Fe atoms on the SD-Fe_1-Ti electrode was also observed (Fig. 1f–h). Moreover, the electrical conductivity of SP-Fe_1-Ti was very close to that of SD-Fe_1-Ti (Supplementary Table 1), benefiting from the excellent electrical conductivity of Ti substrate.

### Experimental evidences for the spin polarization of Fe_1−Ti pairs induced by OVs

The unpaired electrons located in OVs are expected to transfer to the empty 3d levels belonging to Ti and Fe atoms adjacent to OVs[36,37], resulting in two possible consequences including a negative shift in the core level binding energies of the reduced Ti or Fe atoms, and the presence of an unpaired electron (spin) in the 3d shell of the Ti or Fe atoms[38,39]. While the electronic properties of Fe and Ti in SD-Fe_1-Ti electrode without OVs should be different from those of SP-Fe_1-Ti electrode, maintaining spin-depressed $Fe^{3+}$ and $Ti^{4+}$ states. To confirm the electronic structures and spin states of SP-Fe_1-Ti and SD-Fe_1-Ti, ESR, X-ray photoelectron spectroscopy (XPS), soft X-ray absorption spectroscopy (XAS), Mössbauer spectroscopy, and temperature-dependent magnetizations (M-T) measurements were carried out. We first investigated the energy alignment of the Fermi levels ($E_F$, see the calculation details in Supplementary Fig. 11) for TiO_x/Ti, SP-Fe_1-Ti, and Fe (Fig. 2a). It was found that the calculated $E_F$ of TiO_x (~−3.64) was above that of Fe (~−4.5)[40], so the trapped electrons at the OVs on the amorphous TiO_x surface can transfer to the supported Fe atoms, resulting in the $E_F$ upshift of SP-Fe_1-Ti[41]. As seen from the high

−resolution Fe $2p$ core level spectra in Fig. 2b, two peaks at 710.7 and 723.8 eV were observed for SD−Fe$_1$−Ti, which could be assigned to the Fe $2p_{3/2}$ and Fe $2p_{1/2}$ orbitals of trivalent Fe$^{3+}$ species[42,43]. Compared with the Fe $2p$ spectra of SD−Fe$_1$−Ti, the Fe $2p_{3/2}$ and Fe $2p_{1/2}$ peaks of SP−Fe$_1$−Ti showed an obviously lower−energy shift in binding energy, suggesting the reduction of Fe valence state induced by OVs and the appearance of electron−rich Fe$^{2+}$ species. As such, OVs on the amorphous TiO$_x$ surface of SP−Fe$_1$−Ti can transfer electron to the Fe single atoms and lead to an increased electron density of Fe single atoms. Such an electron transfer from OVs to Fe atoms even occurs within

FeCl$_3$/TiO$_x$ without thermal treatment, as evidenced by the appearance of Fe$^{2+}$ species in its Fe $2p$ XPS spectra (Supplementary Fig. 12). We further applied the Mössbauer spectroscopy to reveal the spin state of Fe single atoms on the Ti monolithic electrode (Fig. 2c), and found no sextets or singlets but doublets were derived from the deconvolution of the Mössbauer spectra for SP−Fe$_1$−Ti and SD−Fe$_1$−Ti, indicating the absence of Fe−Fe bonds[44,45]. As expected, SP−Fe$_1$−Ti possessed a much higher content of Fe$^{2+}$ species (40.2%) with high spin polarization configuration (t$_{2g}^4$ e$_g^2$) than SD−Fe$_1$−Ti (26.0%) according to the fitted results (Supplementary Table 2).

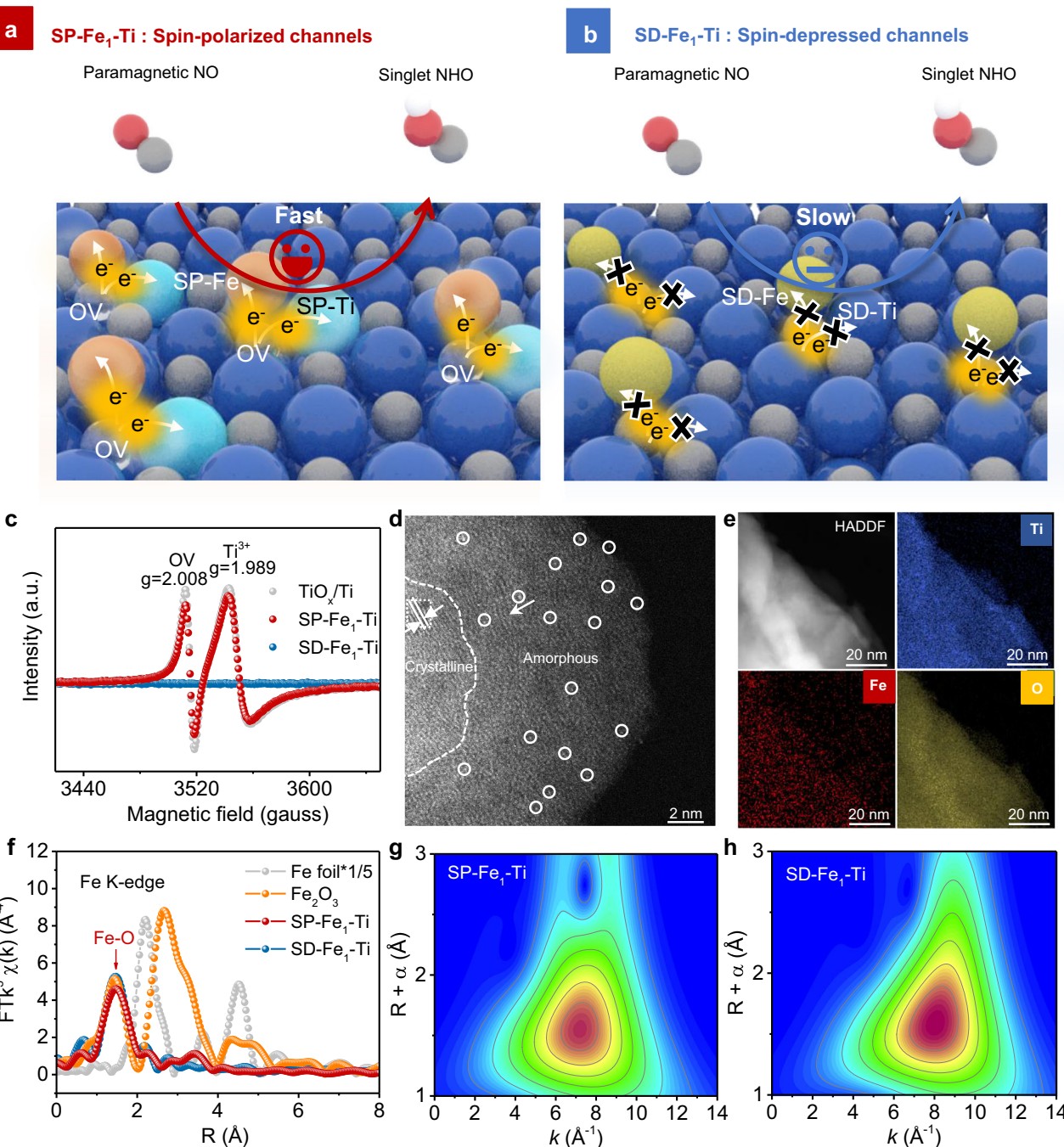

**Fig. 1 | Manipulating spin polarization of Fe$_1$−Ti pairs.** Schematic illustration of the manipulation strategy of spin polarization for Fe$_1$−Ti pairs on inherent oxide surface of Ti foam for **a** spin−polarized Fe$_1$−Ti (SP−Fe$_1$−Ti) and **b** spin−depressed Fe$_1$−Ti (SD −Fe$_1$−Ti). Orange, yellow, cyan, blue, grey and transparent balls represent spin−polarized Fe, spin−depressed Fe, spin−polarized Ti, spin−depressed Ti, oxygen and oxygen vacancy (OV), respectively. **c** ESR spectra of TiO$_x$/Ti, SD−Fe$_1$−Ti and SP −Fe$_1$−Ti. **d** HAADF−STEM image of SP−Fe$_1$−Ti, where Fe single atoms were indicated by a white circle. **e** HADDF image and STEM elemental mapping of SP−Fe$_1$−Ti. **f** FT−EXAFS spectra of SP−Fe$_1$−Ti, SD−Fe$_1$−Ti, Fe foil and Fe$_2$O$_3$ at the Fe K−edge. k$^3$−weighted WT −EXAFS spectra of **g** SP−Fe$_1$−Ti and **h** SD−Fe$_1$−Ti at the Fe K−edge.

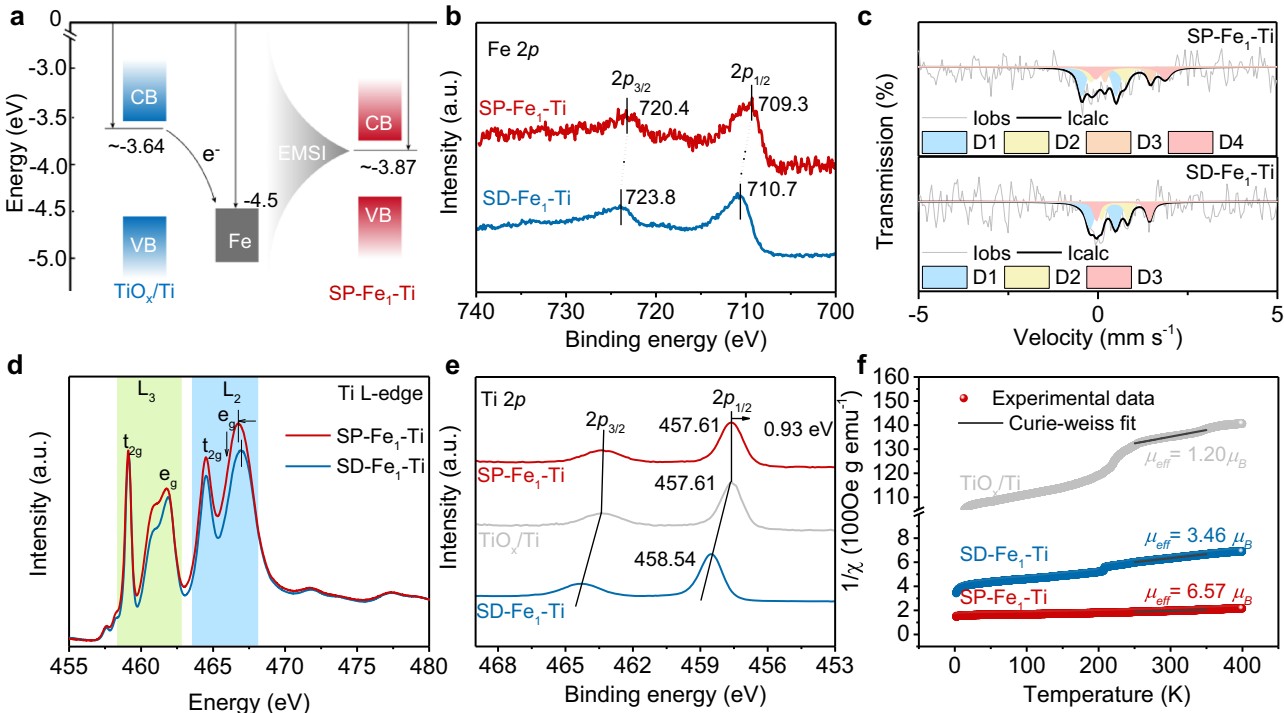

**Fig. 2 | Experimental evidences for the spin polarization of Fe₁−Ti pairs induced by OVs. a** Energy alignment of the $E_f$ for TiOx/Ti, SP−Fe₁−Ti, and Fe foil. **b** Fe 2p XPS spectra of SD−Fe₁−Ti and SP−Fe₁ − Ti. **c** Mössbauer spectra for SP−Fe₁−Ti and SD −Fe₁−Ti. **d** Ti L−edge soft−XAS spectra of SD−Fe₁−Ti and SP−Fe₁−Ti. **e** Ti 2p XPS spectra of SD−Fe₁−Ti and SP−Fe₁−Ti. **f** $1/\chi_m$ versus temperature plots and the calculated $\mu_{eff}$ of TiOx/Ti, SD−Fe₁−Ti and SP−Fe₁−Ti.

Furthermore, we found that OVs on both TiOx/Ti and SP−Fe₁−Ti were accompanied by the appearance of spin−polarized $Ti^{3+}$ species with an EPR signal at $g = 1.989$, which was invisible in the EPR spectra of SD−Fe₁−Ti without OVs (Fig. 1c). This difference indicated that the introduction of OVs could trigger the spin−polarization of Ti sites. We thus employed the Ti L−edge soft−XAS measurement that is highly sensitive to charge state, orbitals occupation and the spin orientation of electrons[46–48] to reveal the formation of spin−polarized $Ti^{3+}$ species on SP−Fe₁−Ti (Fig. 2d). The multiple spectral features of SD−Fe₁−Ti were very similar to that of SrTiO₃ indicating $Ti^{4+}$ valence state[49]. In comparison with SD−Fe₁−Ti, the lower−energy shift and higher L₂ peak intensity of SP−Fe₁−Ti verified the formation of spin−polarized $Ti^{3+}$ species induced by OVs[47], which was further confirmed by the shift to lower binding energy in Ti 2p XPS spectra for both TiOx/Ti and SP −Fe₁−Ti (Fig. 2e).

We therefore carried out the temperature−dependent magnetic susceptibility (M−T) measurement to unravel the total effective magnetic moment ($\mu_{eff}$), which can be determined via the Curie−Weiss law[50,51]. According to the $1/\chi_m$ versus temperature plots of TiOx/Ti, SP −Fe₁−Ti and SD−Fe₁−Ti (Fig. 2f), the order of calculated $\mu_{eff}$ was in the sequence of SP−Fe₁ − Ti ($\mu_{eff} = 6.57$) > SD−Fe₁−Ti ($\mu_{eff} = 3.46$) > TiOx/Ti ($\mu_{eff} = 1.20$), suggesting that SP−Fe₁−Ti possessed much more spin electrons than SD−Fe₁−Ti and TiOx/Ti. Taken together, these above results evidently showed that the spin polarization of atomically dispersed Fe₁−Ti pairs can be easily manipulated by OVs, providing an opportunity for developing unique spin−polarized Fe−based single atom catalysts towards NITRR and deepening insights into spin effect on NITRR.

**Evaluation of NITRR performance towards different electrodes**

NITRR performance was investigated on different electrodes in a H −type electrolytic cell under ambient conditions. For the linear sweep voltammetry (LSV) in 1 M KOH (Fig. 3a), the addition of nitrate much more increased the current density of SP−Fe₁−Ti electrode than those

of Ti foam, TiOx/Ti and SD−Fe₁−Ti electrodes. We further quantified the NH₃ Faradaic efficiency ($FE_{NH3}$) and NH₃ yield rate at a certain potential after 2 h of electrolysis via ultraviolet-visible (UV–Vis) spectrophotometry (Supplementary Figs. 13 and 14). As expected, the SP −Fe₁−Ti electrode exhibited higher $FE_{NH3}$ and NH₃ yield rate than the SD−Fe₁−Ti electrode (Fig. 3b). Impressively, the SP−Fe₁−Ti electrode displayed an attractive NH₃ yield rate of 272000 μg h⁻¹ mgFe⁻¹ (16 molNH3 gFe⁻¹ h⁻¹) and a high Faradaic efficiency of 95.2% at −0.4 V vs. RHE, which were also verified by the results obtained from ¹H nuclear magnetic resonance (NMR, Supplementary Fig. 15) measurement. Such an excellent NITRR activity of SP−Fe₁−Ti electrode was much superior to those of state−of−the−art NITRR electrocatalysts, Haber−Bosch catalysts and N₂ reduction electrocatalysts (Fig. 3c and Supplementary Table 3). Additionally, SP−Fe₁−Ti electrode possesses promising application potential for maximizing $FE_{NH3}$, onset potential and current density at certain potential (Supplementary Table 4) over almost all of top−level NITRR electrodes consisting of the low−price components (e.g., Cu, Ti, and Fe in Supplementary Table 5). We also drop −casted SP−Fe₁−Ti catalyst powders on the carbon fiber paper with the catalyst mass loading of 1 mg cm⁻² for performance assessment. As shown in Supplementary Fig. 16, SP−Fe₁−Ti catalyst displays a high cathodic current density of 190 mA cm⁻² at −0.4 V vs. RHE, superior to most of the top−level NITRR catalysts. The NH₃ yield rate defined by the electrode area of SP−Fe₁−Ti catalyst was determined to be 0.99 mmol cm⁻² h⁻¹, which outperforms most of low−price metal −based catalysts and even several high−price metal−based catalysts (Supplementary Table 6). More importantly, the monolithic nature of SP−Fe₁ − Ti electrode is much more feasible than the powder form of previous top−level NITRR electrocatalysts for practical application.

The contributions of other possible products to the faradaic current were also evaluated using gas chromatography equipped with thermal conductivity detector (TCD) and UV–Vis spectrometry. Only H₂ and NO₂⁻ were detected (Supplementary Fig. 17 and Supplementary Fig. 18), and their FEs were calculated to be 0.7% and 4.1%, respectively,

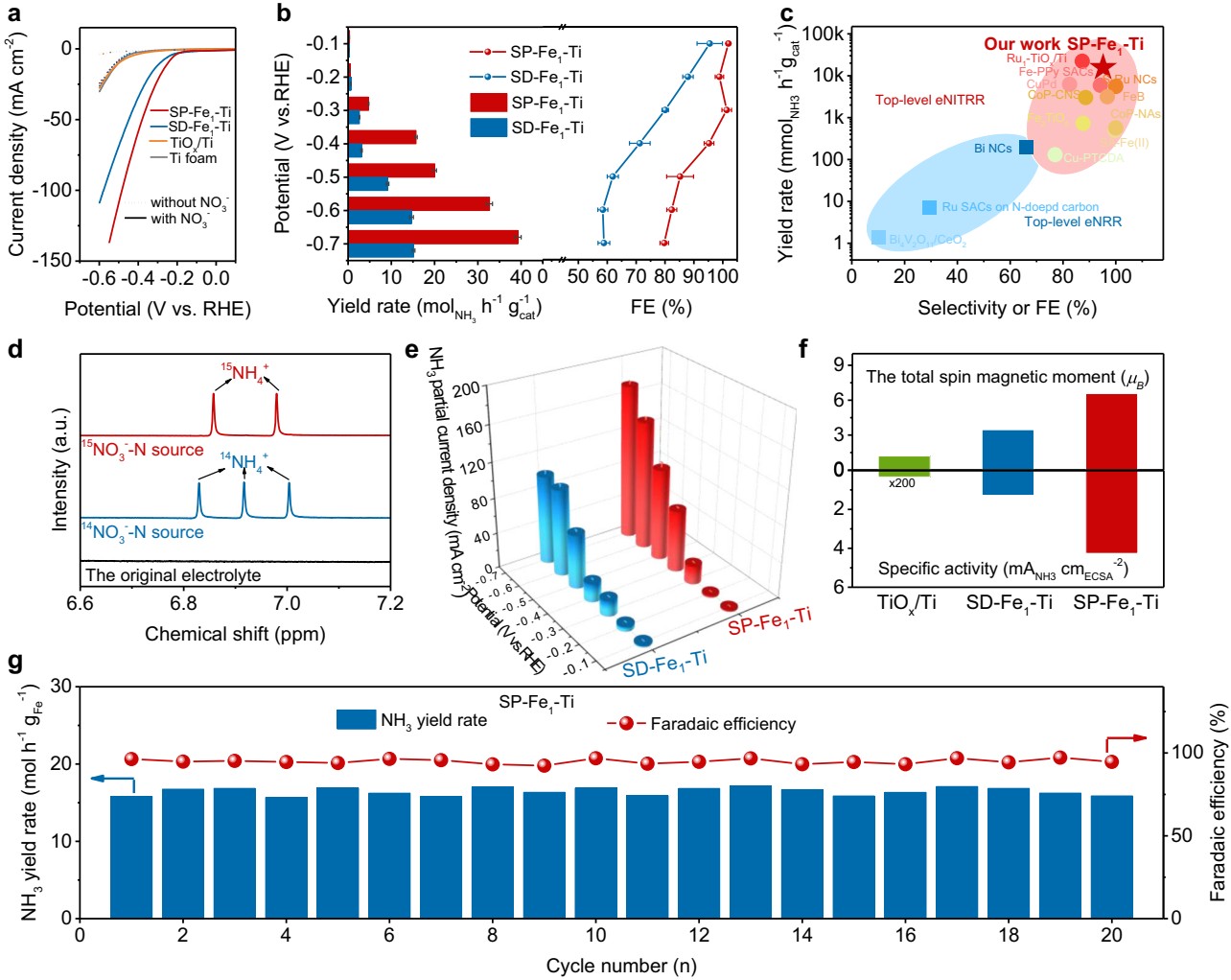

**Fig. 3 | NITRR performances of different electrodes. a** LSV curves of Ti foam, TiO$_x$/Ti, SD–Fe$_1$ – Ti and SP–Fe$_1$ – Ti electrodes. **b** NH$_3$ yield rate and FE$_{NH3}$ of SD –Fe$_1$-Ti and SP–Fe$_1$-Ti electrodes at various potentials. **c** Comparison of the electrocatalytic NITRR performance of SP–Fe$_1$-Ti electrode with other extensively reported electrocatalysts. **d** NMR spectrum of the products generated during the electrocatalytic NITRR on the SP–Fe$_1$–Ti monolithic electrode at −0.4 V vs. RHE. **e** Partial NH$_3$ current densities of SD–Fe$_1$–Ti and SP–Fe$_1$–Ti electrodes normalized to the geometric area at various potentials. **f** Specific activity normalized to ECSA of TiO$_x$/Ti, SD–Fe$_1$ – Ti and SP–Fe$_1$–Ti electrodes at −0.4 V vs. RHE. **g** NH$_3$ yield rate and Faradaic efficiency of SP–Fe$_1$–Ti electrode under the applied potential of −0.4 V vs. RHE during 20 consecutive electrolysis cycles.

at −0.4 V vs. RHE. After taking the two side–reactions into consideration, the total FE (Supplementary Fig. 19) was 100% as expected. Moreover, the NH$_3$ selectivity of SP–Fe$_1$–Ti electrode was determined to be 95.9% at −0.4 V vs. RHE, further confirming its high selectivity towards NH$_3$ production from NITRR.

We thus employed NMR to confirm the N source of generated NH$_3$ via the isotope labeling experiments with $^{15}$N–labeled NO$_3^-$ as the reagent. Different from $^{14}$NH$_4^+$ with three peaks in the $^1$H NMR spectra, only two peaks of $^{15}$NH$_4^+$ appeared in $^1$H NMR spectra when $^{15}$NO$_3^-$ was used as the reagent (Fig. 3d), demonstrating that the produced NH$_3$ was from nitrate feedstock instead of contaminations. In addition, peaks of $^{15}$NH$_4^+$ are absent in $^1$H NMR spectra of the original electrolyte containing $^{15}$NO$_3^-$, further confirming that the produced NH$_3$ was originated from the reduction of nitrate feedstock instead of contaminations. Remarkably, the SP–Fe$_1$ – Ti electrode also showed a larger NH$_3$ partial current density of 174 mA cm$^{-2}$ at −0.7 V vs. RHE than the SD–Fe$_1$–Ti electrode (Fig. 3e), suggesting a great potential in practical applications. We also assessed the intrinsic activity of SP –Fe$_1$–Ti electrode through calculating the specific activity (SA) by normalizing the electrode activity to the electrochemical surface area[52] (ECSA, Supplementary Fig. 20), and found that the SA of SP–Fe$_1$–Ti

electrode was apparently higher than SD–Fe$_1$ – Ti and TiO$_x$/Ti electrodes at −0.4 V vs. RHE (Fig. 3f), revealing the intrinsically high catalytic NITRR activity of spin–polarized Fe$_1$–Ti pairs and the great contribution of spin–polarization of active metal sites to NITRR.

Since the nitrate concentration varies in different sources, we also checked the NITRR activity of SP–Fe$_1$–Ti electrode in the electrolyte with different NO$_3^-$ concentrations (0.1, 0.5, and 1 M) at −0.4 V vs. RHE. As shown in Supplementary Fig. 21, the FEs of NO$_3^-$–to–NH$_3$ conversion were 95.1%, 96.6% and 95.2% in the electrolyte with 0.1, 0.5, and 1 M NO$_3^-$, respectively. These results suggest that the NO$_3^-$ concentration has no obvious impacts on FE$_{NH3}$ of SP–Fe$_1$–Ti electrode, indicating the wide NO$_3^-$ concentration compatibility. In addition, we observed that the NH$_3$ yield rate was enhanced by increasing the NO$_3^-$ concentrations from 0.1 to 1 M because of the accelerated mass transfer[23]. We further evaluated the NITRR activity of the SP–Fe$_1$–Ti electrode at the neutral condition. Impressively, the SP–Fe$_1$–Ti exhibited comparable NH$_3$ yield rate (15 mol h$^{-1}$ g$_{Fe}^{-1}$) and FE (96.1%) to that of at the alkaline condition (Supplementary Fig. 22), further demonstrating its high potential of environmental application.

More importantly, the FE$_{NH3}$ and NH$_3$ yield rate of SP–Fe$_1$–Ti electrode kept very stable during 20 cycles of consecutive electrolysis

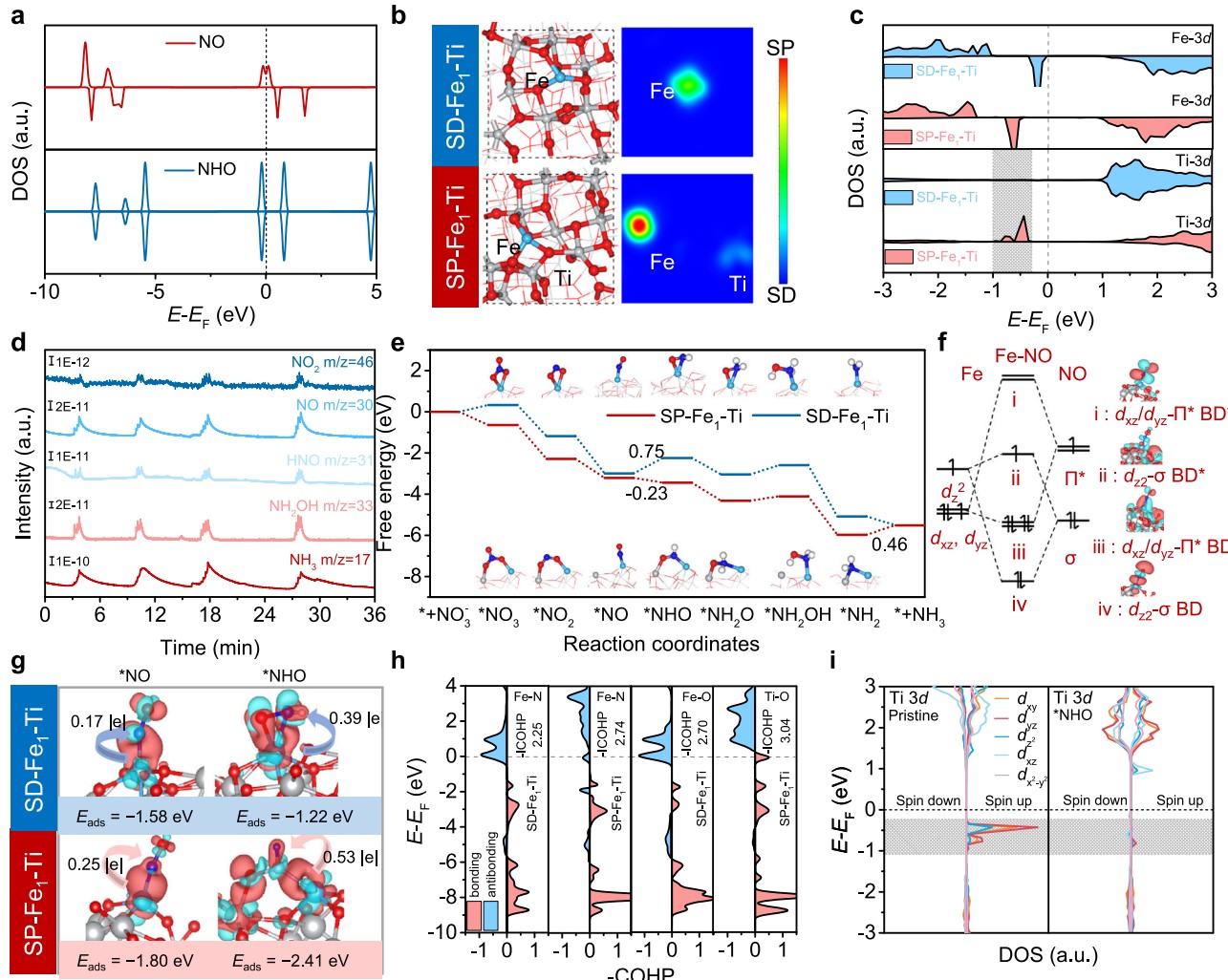

**Fig. 4 | Insights into the NITRR mechanism. a** DOSs of NO and NHO species. **b** Optimized structures and calculated 2D spin density diagrams of SD–Fe$_1$–Ti and SP–Fe$_1$–Ti. **c** DOSs of Fe and Ti atoms on SD–Fe$_1$–Ti and SP–Fe$_1$–Ti. **d** DEMS measurements of NITRR over SP–Fe$_1$–Ti electrode. **e** Calculated free energy diagrams for NITRR on SD–Fe$_1$–Ti and SP–Fe$_1$–Ti. **f** Major orbital interactions between Fe atom and NO molecule as well as the corresponding molecular orbital diagrams (BD and BD* represent bonding and antibonding orbital, respectively). **g** Charge density differences of NO and NHO adsorbed on SD–Fe$_1$–Ti and SP–Fe$_1$–Ti (red and cyan represent electron accumulation and depletion, respectively). **h** COHPs of NHO adsorbed on SD–Fe$_1$–Ti and SP–Fe$_1$–Ti. **i** DOSs of Ti atom on SP–Fe$_1$–Ti before and after NHO adsorption. Ti, O, Fe, N, and H atoms are denoted by grey, red, cyan, blue, and white, respectively.

in a H–type electrolytic cell at −0.4 V vs. RHE (Fig. 3g and Supplementary Fig. 23), indicating its long–term durability and high potential for practical application. XRD, XPS, and TEM analyses further revealed the high stability of SP–Fe$_1$–Ti electrode during NITRR (Supplementary Figs. 24–26), as confirmed by the negligible Fe dissolution of SP–Fe$_1$–Ti electrode within 20 cycles consecutive electrolysis (Supplementary Fig. 27).

**Insights into NITRR mechanism towards SP–Fe$_1$–Ti pairs**

Density functional theory (DFT) calculations were employed to get insights into NITRR mechanism over SP–Fe$_1$–Ti pairs. We first explored the density of states (DOSs) of the key NO and NHO species involved in NITRR (Fig. 4a), and confirmed the asymmetric spin state of NO in DOS of spin–up and spin–down states, which was different from that of NHO, indicating that the transformation of NO to NHO was accompanied by a spin–state transition. As revealed by the constructed models of SP –Fe$_1$–Ti and SD–Fe$_1$–Ti (Fig. 4b), their Fe atoms were respectively coordinated with three and four O atoms for SP–Fe$_1$–Ti and SD–Fe$_1$–Ti, consistent well with the fitting EXAFS results (Supplementary Fig. 28 and Table 7). The spin density diagrams displayed that only Fe atom was spin–polarized for SD–Fe$_1$–Ti, while both Fe atom and Ti atom adjacent

to Fe atom were spin–polarized in the case of SP–Fe$_1$–Ti (Fig. 4b). This magnetic difference further verified the increase of the M–T measured magnetic moment with the existence of OV (Fig. 2f). Magnetic states were also evident from the DOSs of Fe and Ti atoms. The spin up and spin down components of Fe atom around Fermi level ($E_F$) were quite different (Fig. 4c), indicative of unpaired electrons in the Fe–3d orbitals. Besides, the spin–up and spin–down components of Ti atom in SD –Fe$_1$–Ti were roughly the same, but unpaired electronic states appeared near $E_F$ after the introduction of OV (Fig. 4c). These magnetic Ti and Fe atoms contributed to the highly efficient NITRR.

Given that the hydrogen source for NITRR under alkaline condition comes from H$_2$O, water dissociation including the HO–H bond cleavage and the proton transfer ability would strongly influence the catalytic activity of NITRR. It was found that H$_2$O dissociation was slightly endothermic by 0.08 and 0.03 eV, with barriers of 0.28 and 0.29 eV on SD–Fe$_1$–Ti and SP–Fe$_1$–Ti, respectively, suggesting that H$_2$O dissociation on the both electrodes were much favorable and comparable (Supplementary Fig. 29). We further employed the electron spin resonance (ESR) technique using 5,5–dimethyl–1–pyrroline–N –oxide (DMPO) as the radical trapping reagent to investigate the generation and the role of hydrogen radicals from water dissociation

to understand the mechanism of nitrate reduction. Nine ESR peaks with an intensity ratio of 1:1:2:1:2:1:2:1:1 were observed in Supplementary Fig. 30 for the electrocatalysis on the SP−Fe$_1$−Ti electrode in pure 1 mol L$^{-1}$ KOH, which could be assigned to the spin adduct of DMPO−H, confirming the generation of H$_{ads}$. When adding NaNO$_3$ in 1 mol L$^{-1}$ KOH, the signal intensity of DMPO−H disappeared. Such a phenomenon revealed that the generated H$_{ads}$ through water dissociation was consumed, suggesting the deep participation of H* in the hydrogeneration process of NITRR. We further conducted kinetic isotope effect (KIE) experiments to estimate the proton transfer ability of SP−Fe$_1$−Ti and SD−Fe$_1$−Ti during NITRR by comparing the current density ratios. The NITRR KIE value (1.40) of SP−Fe$_1$−Ti was close to that (1.53) of SD−Fe$_1$−Ti at −0.4 V vs. RHE (Supplementary Fig. 31), suggesting a similar hydrogen transfer ability on the two electrodes. Therefore, the water dissociation and hydrogen transfer abilities have tiny impact on the distinct NITRR activities of SP−Fe$_1$−Ti and SD−Fe$_1$−Ti.

To understand the unique role of spin−polarized Fe$_1$-Ti pairs on NITRR, the NITRR reaction pathways on SD−Fe$_1$−Ti and SP−Fe$_1$−Ti were investigated. To construct a comprehensive description of the reaction mechanism, online differential electrochemical mass spectrometry (DEMS) was utilized to on-line detect the key intermediates and products (Fig. 4d). We observed the m/z signals of 46, 30, 31, 33, and 17 from NO$_2$, NO, NHO, NH$_2$OH, and NH$_3$, respectively. On the basis of DEMS results, we calculated the free energy of individual intermediate on SD−Fe$_1$−Ti and SP−Fe$_1$−Ti (Fig. 4e, Supplementary Fig. 32) under a potential of 0 V vs. RHE[53–55]. Actually, NO$_3^-$ to *NO underwent an NO$_3^-$ adsorption step and two deoxygenation steps, namely, *+NO$_3^-$ → *NO$_3$ → *NO$_2$ → *NO. In the case of the SD−Fe$_1$−Ti system, the NO$_3^-$ adsorption underwent an uphill Gibbs free energy change of 0.34 eV. Afterward, *NO$_3$ first deoxygenated to *NO$_2$ with a downhill free energy change of −1.52 eV. Subsequently, *NO$_2$ continued to remove one oxygen to form *NO with an energy release of 1.81 eV. As for the SP−Fe$_1$−Ti system, the above three steps release the energy of 0.64, 1.65, and 0.93 eV, respectively. Finally, NO$_3^-$ deoxygenated to *NO with a total downhill Gibbs free energy change of −3.00 eV on SD−Fe$_1$−Ti, while the total downhill free energy change for the above steps was −3.22 eV on SP−Fe$_1$−Ti. Therefore, NO$_3^-$ deoxygenation to *NO on SP−Fe$_1$−Ti was more favorable thermodynamically. For NITRR on SD−Fe$_1$−Ti, the potential−determining step (PDS) was *NO to *NHO with an thermodynamic barrier of 0.75 eV, while the PDS was determined to be *NH$_2$ to NH$_3$ with a thermodynamic barrier of 0.46 eV on SP−Fe$_1$−Ti, consequently resulting in a higher NITRR activity than SD−Fe$_1$−Ti[53,55,56]. This PDS change stemmed from the effect of both spin−polarized Fe and Ti sites on SP−Fe$_1$−Ti, resulting in the remarkably decreased free energy of *NO to *NHO from 0.98 eV to −0.23 eV. Meanwhile, a much smaller peak intensity ratio (-3.92) of *NO to *NHO for SP−Fe$_1$−Ti than that of SD−Fe$_1$−Ti (-5.83, Supplementary Fig. 33) was observed, further confirming an accelerated *NO hydrogenation process on the SP−Fe$_1$−Ti electrode. To further highlight the synergistic effect of Fe$_1$−Ti spin pairs, we calculated the free energy of *NHO intermediate adsorbed on the Fe$_1$−Ti pairs involving a Ti atom without spin−polarization for comparison (Supplementary Fig. 34), and the calculated value of 0.37 eV was much higher than that on SP−Fe$_1$ − Ti, suggesting that the spin−polarization of Ti atom was also significantly contributed the higher activity of NITRR. Subsequently, we investigated the competitive hydrogen evolution reaction on spin−polarized Fe$_1$ − Ti pairs (Supplementary Fig. 35), and the calculated free energy for *H adsorption on Fe and Ti sites of SP−Fe$_1$−Ti were 1.13 and 0.78 eV, respectively, much weaker than that of NO$_3^-$ adsorbed on spin−polarized Fe$_1$−Ti pairs (−0.64 eV). Therefore, the H$_2$ generation on the spin−polarized Fe$_1$−Ti pairs were inhibited, facilitating to enhance the Faradaic efficiency for NITRR.

To further understand the spin−polarization effect of Fe$_1$−Ti pairs on promoting the activity of NITRR, the orbital interactions between *NO/*NHO and the Fe-Ti pair were analyzed in detail, since *NO

hydrogenation was generally suggested to govern the activity[24]. For NO adsorption on SP−Fe$_1$−Ti, it was found that NO preferably adsorbed on Fe (−1.80 eV), stronger than that on Ti (−1.09 eV). The main contribution to the interaction between Fe and *NO was from the d$_{xz}$/d$_{yz}$/d$_{z^2}$ orbitals of Fe atom and the σ/π* orbitals of NO (Fig. 4f), according to the high spin polarization configuration (t$_{2g}^4$ e$_g^2$) of Fe, as mentioned above. We noted that the d$_{xy}$ and d$_{x^2-y^2}$ orbitals of Fe could not hybridize with the NO orbital owing to symmetry conservation, their orbital interaction was thus not considered[57,58]. Thus, in such a bonding model, the semioccupied d$_{z^2}$ of Fe can accept electrons from the occupied σ orbital of NO, and in turn the electrons of Fe d$_{xz}$/d$_{yz}$ orbitals can inject into the partially occupied π* orbitals to weaken the bond order of NO[57,59,60]. This "donation−backdonation" of electrons generally led to the NO's acceptance of 0.25 |e| from the substrate (Fig. 4g and Supplementary Fig. 36), resulting in a binding energy of −1.80 eV for NO adsorption on the Fe site of SP−Fe$_1$−Ti, stronger than that of −1.58 eV on the Fe site of SD−Fe$_1$−Ti that transferred less electron of 0.17 |e| to *NO. Moreover, the spin polarized pairs can also be advantageous of the hydrogenation of *NO through significantly stabilizing hydrogenated *NHO intermediates. The calculated adsorption energy of −2.41 eV for *NHO that preferably adsorbed on SP−Fe$_1$−Ti via the formation of Fe−N and Ti−O was much stronger than that of −1.22 eV on SD−Fe$_1$−Ti via forming Fe−N and Fe−O bonds, because more electrons transferred to *NHO from SP−Fe$_1$−Ti (0.53 |e|) than that from SD−Fe$_1$−Ti (0.39 |e|), as evidenced by the Bader charge analysis (Fig. 4g), which was supported by the Crystal Orbital Hamiltonian Populations (COHP) analysis (Fig. 4h). The negatively integrated COHP (−COHP) values for Fe−N (2.74 eV) and Ti−O (3.04 eV) interactions on SP−Fe$_1$−Ti were more positive than those for Fe−N (2.25 eV) and Fe−O (2.70 eV) interactions on SD−Fe$_1$−Ti, suggesting that the spin−polarized Fe$_1$−Ti pairs exhibited a stronger interaction with *NHO. As stated before, Ti atom adjacent to Fe atom of SP−Fe$_1$−Ti underwent a spin state transition from unspin polarization to spin polarization after the introduction of OVs, as supported by the appearance of unpaired d$_{yz}$, d$_{xy}$, and d$_{z^2}$ states near the $E_F$ (Fig. 5i). Furthermore, the d$_{z^2}$ spin state near the $E_F$ of Ti atom on SP−Fe$_1$−Ti disappeared after the adsorption of *NHO, implying that the unpaired spin electron of Ti atom injected into *NHO[61,62]. Similarly, the unpaired spin electron in d$_{z^2}$ state of Fe can also partially inject into *NHO (Supplementary Fig. 37), enhancing the electron transfer from Fe−Ti spin pair to further stabilize *NHO. Conclusively, both spin−polarized Fe−Ti pair sites can effectively facilitate the deoxygenation of NO$_3^-$ to *NO and stabilize *NO and *NHO intermediates to boost the subsequent *NO hydrogenation, as well as inhibit the hydrogen evolution reaction, leading to high activity and selectivity of SP−Fe$_1$−Ti electrode towards NITRR.

## An integrated equipment for flow−through NITRR electrolyzer and in−situ ammonia recovery

Besides the electrode performance, NH$_3$ product separation and recovery are also crucial for the practical application of NITRR[11,63,64]. From this aspect, we designed an integrated device composed of a flow−through NITRR electrolyzer and a membrane−based ammonia recovery unit to realize the efficient NITRR and in-situ NH$_3$ recovery, as schematically illustrated in Fig. 5a and Supplementary Fig. 38. We designed a compact flow−through electrolyzer consisting of SP−Fe$_1$−Ti cathode and the mesh-type commercial dimension stable anode (DSA, the Ru$_{0.8}$Ir$_{0.2}$O$_2$/Ti electrodes with a coating thickness of -10 μm) as the NITRR electrolyzer. During operation, the NO$_3^-$−containing solution (250 mL) as synthetic effluent was treated through electrochemical reduction on SP−Fe$_1$−Ti cathode, and circulated by a peristaltic pump with a flow rate of 60 mL min$^{-1}$. Then the NH$_3$−containing effluent was directly guided into the hollow polypropylene (PP) fiber arrays immersed into 1 M HCl solution (800 mL) to recover NH$_3$. The key of simultaneous NITRR and NH$_3$ product recovery is the continuous flow of NH$_3$ through the gas−permeable hydrophobic membrane and then

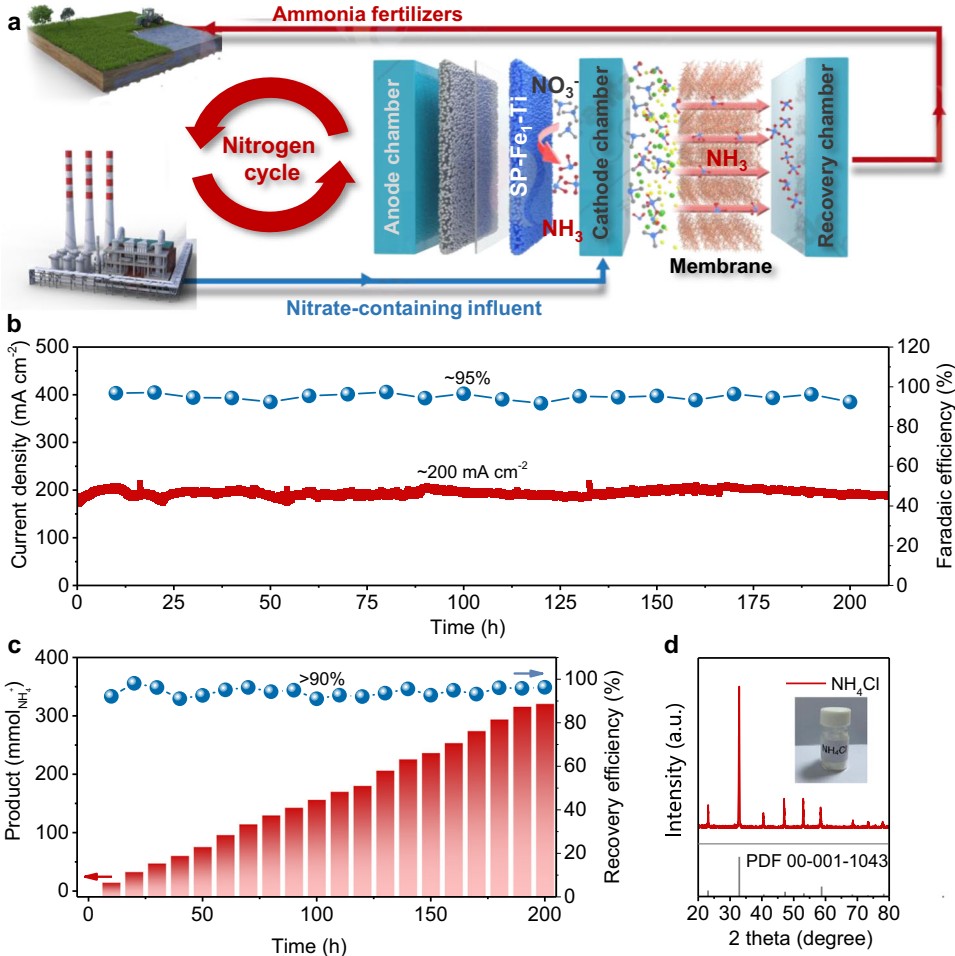

**Fig. 5 | An integrated equipment for flow−through NITRR electrolyzer and in −situ NH₃ recovery. a** Schematic illustration of integrated equipment composed of flow−cell reactor and membrane separator by employing the SP−Fe₁−Ti monolithic cathode for NITRR and a gas−permeable hydrophobic membrane for NH₃ recovery. **b** Long−term stability test of SP−Fe₁−Ti electrode at −0.4 V vs. RHE using the integrated equipment. **c** NH₃ production and recovery efficiency of the integrated equipment during long−term stability test. **d** XRD pattern of the obtained NH₄Cl powder.

back to the cathode chamber. During this process, the produced NH₃ could permeate through membrane between the cathode chamber and the recovery chamber and be absorbed by acid solution in the recovery chamber[63,64]. By using the SP−Fe₁−Ti monolithic cathode, the coupled device could robustly work at an industrial−level current density of ~200 mA cm⁻² for up to 200 h with a high $FE_{NH_3}$ of 95% (Fig. 5b). As a result, this sustainable and decentralized system achieved high NITRR and simultaneous NH₃ recovery with nearly 100% selectivity and about 90% recovery efficiency (Fig. 5c). After long−term electrolysis, the measured contact angle of membrane (Supplementary Fig. 39) was determined to be 123.5°, similar with that of pristine membrane (123.5°), suggesting that the post−reacted membrane still kept its hydrophobicity. After further rotary evaporation, the recycled ammonia in solution could be converted into high−purity NH₄Cl powder[11], as confirmed by the XRD pattern (Fig. 5d). These findings demonstrated the feasibility of selective conversion of nitrate−containing wastewater into upgraded ammonia fertilizers through combining the high−performance SP−Fe₁−Ti monolithic cathode with the integrated device, providing a sustainable way for nitrogen cycle.

## Discussion

In conclusion, the spin−polarized Fe₁−Ti pairs were designed by manipulating oxygen vacancies of the inherent surface oxide layer on monolithic Ti electrode, and the resultant electrode could deliver extraordinary NITRR activity with an impressive NH₃ yield rate

(272,000 µg h⁻¹ mg$_{Fe}$⁻¹) and a high NH₃ Faradic efficiency of 95.2% at −0.4 V vs. RHE, suppressing the counterpart of spin−depressed Fe₁−Ti electrode and most of the well−known electrodes. It was demonstrated that spin−polarized Fe−Ti pairs could provide spin electrons to interact with the reaction intermediates, and consequently facilitate to the deoxygenation of NO₃⁻ to *NO and the hydrogenation of *NO via strengthening their adsorption. By coupling the NITRR flow−through electrolyzer with the membrane separation technology, we successfully realized the on−site ammonia recovery in the fertilizer's formation from the nitrate−containing electrolyte in a sustainable and decentralized manner, by taking advantage of the resultant spin−polarized Fe₁−Ti pairs electrode. Our study not only showcases the construction of spin−polarized Fe₁−Ti pairs for NITRR and potential wastewater treatment or ammonia production applications, but also offers an innovative strategy to develop the advanced electrodes with spin polarization effect towards spin−related reactions.

## Methods
### Chemicals
The titanium monolithic electrode was purchased from Kunshan Guangjiayuan New Material Co., Ltd., China. Iron chloride (FeCl₃·6H₂O), potassium sodium tartrate (KNaC₄H₆O₆), DMSO−d6, maleic acid (C₄H₄O₄), sodium nitrate (NaNO₃), potassium hydroxide (KOH), potassium nitrate (K¹⁵NO₃) and ethanol were purchased from

Sinopharm Chemical Regent Company. All chemicals were used without further purification.

## Electrode preparation

The synthesis route of SP–Fe₁–Ti electrode was illustrated in Supplementary Fig. 1a, as previously reported[1]. Typically, the TiOₓ/Ti monolithic electrode with 2 cm × 2 cm in size was first pretreated in reductive atmosphere (5 % H₂/Ar) at 400 °C for 3 h with a heating rate of 5 °C min⁻¹ to create the oxygen vacancies (OVs) on the electrode surface. Then, the homogeneous Fe precursor solution with a Fe mass concentration of 4 mg mL⁻¹ in ethanol was uniformly deposited onto the as-prepared TiOₓ/Ti monolithic electrode. Meanwhile, the deposition process was under the infrared lamp illuminating to fasten the solvent evaporation and avoid the solvent aggregation arising from surface tension. Finally, the SP–Fe₁–Ti electrode was obtained by treating the Fe-deposited electrode in reductive atmosphere (5 % H₂/Ar) at 400 °C for 3 h with a heating rate of 5 °C min⁻¹ once more. The SD–Fe₁–Ti electrode followed the similar synthesis route but in an oxidative atmosphere (O₂/Air) to avoid the generation of OVs for comparison (Supplementary Fig. 1b).

## Characterization

X-ray diffraction (XRD) patterns were recorded by employing a Rigaku Miniflex-600 (Cu Kα, λ = 0.15406 nm, 40 kV and 15 mA). The high-angle annular dark-field scanning transmission electron microscope (HAADF–STEM) images were obtained on a JEOL JEM–ARM200F TEM/STEM with a spherical aberration corrector working at 200 kV. X-ray photoelectron spectroscopy (XPS) was carried out on scanning X-ray microprobe (PHI 5000 Verasa, ULAC–PHI, Inc.) with Al Ka radiation. The C 1 s peak at 284.6 eV is as internal standard for energy calibration. The energy dispersive X-ray spectroscopy (EDS) was performed on FEI Talos F200X. Electron spin resonance (ESR) was conducted on the EMX micro-6/1 under 100 K. The X-ray absorption fine structure spectra of Fe K-edge was obtained at Singapore Synchrotron Light Source center (SSLS, operating at 2.5 GeV with a maximum current of 200 mA). The EXAFS spectra were determined via subtracting the post-edge background from the overall absorption and then normalized to the edge-jump step. Subsequently, the χ(k) data was Fourier transformed to real (R) space by using a hanning windows (dk = 1.0 Å⁻¹) to separate the EXAFS contributions from different coordination shells. To obtain the quantitative structural parameters around central atoms, least-squares curve parameter fitting was performed with using the ARTEMIS module of IFEFFIT software packages. XAS spectra of Ti–L edge were collected at the BL 11 A beamline of the National Synchrotron Radiation Research Center (NSRRC) in Taiwan. We measured electrical conductivities of SP–Fe₁–Ti and SD–Fe₁–Ti electrode at room temperature through the four probe DC technique.

## Electrochemical measurements

NITRR test was conducted in a typical H–type cell, separated by a Nafion 117 membrane. The Fe₁–Ti electrodes with specific spin polarization degree (SP–Fe₁–Ti and SD–Fe₁–Ti), platinum plate and Hg/HgO electrode were used as working electrode, counter electrode and reference electrode, respectively. If not specified otherwise, all potentials reported in this work were referenced to a reversible hydrogen electrode (RHE). Before each measurement, the electrolyte was purged with Ar (99.99%) for 30 min. The linear sweep voltammetry (LSV) curves were recorded at a scan rate of 10 mV s⁻¹ under stirring with rotation rate of 500 rpm. Potentiostatic tests were conducted at certain applied potentials within NITRR window for 2 h. Nessler's reagent method was employed to quantify the generated NH₃ in electrolyte to further determine the faradaic efficiency (FE_NH₃) and NH₃ yield rate. In particular, a certain amount of electrolyte was taken out from the cell and diluted to the detection range. Then, an aqueous solution of potassium sodium tartrate (KNaC₄H₄O₆, 100 μL, 500 g L⁻¹) and Nessler's reagent

(100 μL) were added into the diluted electrolyte (5 mL) to obtain the uniform mixture. Let stand for 10 min. The UV–Vis absorption intensity at the absorbance peak at 420 nm was collected to determine the NH₃ concentration of a series of standard ammonium chloride solutions and the mixed solution. The concentration–absorbance curve was plotted for the calculation of the NH₃ concentration of the mixed solution (Supplementary Fig. 14). The NH₃ yield rate was calculated via the following relation:

$$r_{NH3} = (c_{NH3} \times V)/(t \times m_{cat}) \qquad (1)$$

The FE_NH₃ was calculated via the following relation:

$$FE_{NH3} = (8 \times F \times c_{NH3} \times V)/(17 \times Q) \qquad (2)$$

where $c_{NH3}$ is the measured NH₃ concentration, V is the volume of the electrolyte, t is the reaction time, $m_{cat}$ is the Fe mass on the monolithic electrode, F is the Faraday constant (96485 C mol⁻¹) and Q is the total charge. The NH₃ amount was further determined by 1H NMR (600 MHz, Bruker Avance III) with 10 vol% DMSO-d6 as the spin-lock field and 0.4 mg L⁻¹ maleic acid (C₄H₄O₄) as the internal standard, respectively. Before each 1H NMR measurement, the pH of obtained solution was adjusted to 2.0 with HCl. The concentration of NH₄⁺ was plotted versus the peak area ratio of NH₄⁺ and C₄H₄O₄ to obtain the standard curve. The generated NH₄⁺ concentration can be determined from standard curve. An isotope-labeling experiment was performed in the electrolyte containing 1 mol L⁻¹ KOH and 1 mol L⁻¹ K¹⁵NO₃/ K¹⁴NO₃ for 2 h at −0.4 V vs. RHE to specify the nitrogen source of NH₃. The obtained ¹⁵NH₄⁺/¹⁴NH₄⁺ was observed in 1H NMR spectra. The possible gas products were detected by a gas chromatography (GC, Thermal Trace-1300) equipped with thermal conductivity detector (TCD). The Griess test was used to determine the NO₂⁻ concentration in electrolytes after the potentiostatic test. Generally, N-1-naphthyl ethylenediamine dihydrochloride (0.04 g), p-aminobenzene sulfonamide (0.8 g), and H₃PO₄ (2 mL, 85%) were dissolved in DI water (10 mL) using the Griess reagent. The diluted electrolyte (5 mL) was mixed with the Griess agent (100 uL) and rested for 10 min at room temperature to conduct UV–vis tests. NO₂⁻ concentrations of the electrolyte were determined by the absorbance at 540 nm. In the same operation, various NaNO₂ aqueous solutions were used as the standard samples to obtain the calibration curve (Supplementary Fig. 17). The *FE* of byproduct *i* (*FEᵢ*) was calculated via the following relation:

$$FE_i = (n_i \times F \times c_i \times V)/(17 \times Q) \qquad (3)$$

The selectivity of NH₃ was calculated via the following relation:

$$S_{NH3} = c_{NH3}/(c_{NH3} + c_{NO2-}) \qquad (4)$$

where $n_i$ is the number of electrons transferred to byproduct *i*, $c_i$ is the measured byproduct *i* concentration, V is the volume of the electrolyte or the upper space, F is the Faraday constant (96485 C mol⁻¹) and Q is the total charge.

The electrical double layer capacitor (C_dl) was determined from double-layer charging curves using cyclic voltammograms (CVs) within non-faradaic potential region between 0.4 and 0.5 V vs. RHE at different scan rates from 20 mV s⁻¹ to 100 mV s⁻¹. Half of the current density difference (ΔJ/2) at the centered potential was plotted against the scan rate (v) to calculate the slope of fitted straight line and obtained the C_dl value. The electrochemical active surface area (ECSA) was calculated by dividing the double-layer capacitance (C_dl) by specific capacitance (C_s, 40 μF cm⁻²). Differential Electrochemical Mass Spectrometry (DEMS) measurements were carried out on a specially-made electrochemical cell with electrolyte constantly flowing in through a peristaltic pump. During the test, Ar was kept bubbling

into the electrolyte. The mass signals were constantly collected during the LSV measurements from 0.1 to −0.6 V at a scan rate of 10 mV s$^{-1}$. When the mass signal returned to baseline, another three cycles were conducted under identical condition to avoid the accidental error.

## Data availability

All data that support the findings of this study are present in the paper and the Supplementary Information. Further information can be acquired from the corresponding authors upon reasonable request.

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

## Acknowledgements

This work was supported by the National Key Research and Development Program of China (2021YFA1201701, 2022YFA150470, L.Z.), the National Natural Science Foundation of China (U22A20402, U21A20286, L.Z.), (22102100, Y.Y.), (22206121, J.D.), (22273068, U22A20394, X.G.), Shenzhen Science and Technology Program (JCYJ20220818095601002, L.Z.), the Natural Science Foundation of Shanghai (22ZR1431700, Y.Y.), the China Postdoctoral Science Foundation (2021M702117, 2022M722080, J.D.), the National Postdoctoral Program for Innovative Talents (BX20220197, J.D.) and the Shanghai Postdoctoral Excellence Program (2021182, J.D.), (2022381, G.Z.). The authors acknowledge the support from the Max Planck–POSTECH–Hsinchu Center for Complex Phase Materials, the Instrumental analysis center of Shanghai Jiao Tong University, Instrumental analysis center of School of Environmental science and engineering, Shiyanjia Lab and Bruker Corporation for the help in characterizations and experimental measurements, and Supercomputing Center of Wuhan University.

## Author contributions

Y. Y., X. G., and L.Z. Z. conceived the idea. J. D and L. Z. carried out the experiments. Y. T. carried out the DFT calculations. J. D., Y. T., Y. Y., X. G., and L.Z. Z. wrote the paper. Z. H., C. C., C. K., G. Z., J. W., X. Z., Q. Z., W. H., R. W., K. W., and R. Z. helped with data analysis and manuscript polishing. All the authors discussed results and provided comments during the manuscript preparation.

## Competing interests

The authors declare no competing interests.
