## [Peer Review File · Nature Communications]

REVIEWER COMMENTS

Reviewer #1 (Remarks to the Author):

Comments on NCOMMS-23-22388

In this work, the authors successfully designed the spin-polarized Fe₁-Ti pairs by manipulating oxygen vacancies of the inherent surface oxide layer on monolithic Ti electrode for electrochemical nitrate reduction to ammonia (NITRR). The resultant electrode with spin-polarized Fe₁-Ti pairs could deliver extraordinary NITRR activity with an unprecedented NH₃ yield rate (272000 μg h⁻¹ mgcat⁻¹) and NH₃ Faradic efficiency (nearly 100 %) at -0.4 V vs. RHE, suppressing the counterpart of spin-depressed Fe₁-Ti electrode and most of the best-known electrodes. They clearly demonstrated that spin-polarized Fe-Ti pairs could provide spin electrons to interact with the reaction intermediates, thus facilitating the deoxygenation of NO₃⁻ to *NO and the hydrogenation of *NO via strengthening their adsorption. Moreover, the in-situ ammonia recovery with high recovery efficiency in this work further ensured the feasibility of the nitrate-to-ammonia conversion process from both energy and environment perspectives. This study provided a new diagram to develop the advanced electrodes with spin polarization effect towards spin-related reactions, which is very attractive for many applications. In addition, the analysis and experiments in this work are quite sound, and the manuscript is carefully organized and well written. We believe this work is definitely suitable for publication in Nature communications after some minor revisions. Some detailed comments are as below.

1. To further demonstrate the application prospect of the electrode with spin-polarized Fe₁-Ti pairs, the authors are suggested to evaluate the catalytic activity of the electrode at the neutral condition.
2. Regarding that the concentrations of NO₃⁻ in various waters are different, it is also necessary to check the impact of NO₃⁻ concentration on the electrode activity.
3. It was widely reported that the hydrogen radical contributed to the nitrate reduction, so the authors should investigate the generation and the roles of hydrogen radicals to understand the mechanism of nitrate reduction with the electron spin resonance (ESR) technique.
4. Please provide the structural/electronic information of electrode and its Fe leaching concentration during the reaction to check the stability of electrode.
5. ¹H NMR spectra of the original electrolyte containing 15NO₃⁻ are required to confirm that the produced NH₃ was originated from the nitrate feedstock instead of contaminations.

Reviewer #2 (Remarks to the Author):

This paper reported a catalyst based on Fe single atom-modified TiO_x/Ti substrate for NO₃⁻-to-NH₃ conversion (NITRR). They suggested that the oxygen vacancies in the TiO_x/Ti substrate could activate the Fe sites and produce spin-polarized Fe¹-Ti pairs for enhanced NITRR performance, compared to the counterpart that underwent heat treatment under the Air/O₂ atmosphere. Nevertheless, the activity of the catalysts did not reach the “superior” state they claimed, and there is insufficient experimental evidence to establish a direct correspondence between spin state changes and activity. More detailed comments are below:

1. In this paper, the authors stated that “The designed spin-polarized Fe¹-Ti pairs delivered an unprecedented NH₃ Faradic efficiency of nearly 100% and an ultrahigh NH₃ yield rate of 272000 μg h⁻¹ mg cat⁻¹ at -0.4 V vs. RHE for NITRR, far superior to the counterpart monolithic electrode with spin-depressed Fe¹-Ti pairs (51000 μg h⁻¹ mgcat⁻¹) and one order of magnitude higher than mostly reported NITRR electrocatalysts.”

Now, numerous catalysts achieve a nearly 100% NH₃ Faradic efficiency even at more positive applied potentials (>-0.2 V vs. RHE). Besides, it is not reasonable to define the NH₃ yield rate by the mass of the Fe single atoms in this catalyst because 1) the Ti sites might take part in the catalytic progress as suggested by the DFT calculations; 2) the price of Ti substrate is much higher than that of the Fe.

On the other hand, the performance of the SP-Fe¹-Ti is only average, as evidenced by the negative onset potentials (~-0.1 to -0.2 V) and a low current density of ~75 mA cm⁻² in electrolytes even with 1 M NO₃⁻ at -0.4 V. As known, the reported catalysts related to the CuRu, CoRu, CuCo show a much better performance than this SP-Fe¹-Ti catalyst.

For a further efficient comparison with other newly reported catalysts, it is necessary to provide the NH₃ yield rate defined by the electrode area, or the sum mass of the electrode (or the high-price components) (the ways used in industry). Note that the progress of an electrocatalytic topic requires a reasonable way to define the real performance of the reported catalysts. It doesn't make sense to get an "eye-catching" number in a biased way.

2. In the introduction, “While the acceleration of spin-transition related *NO hydrogenation via properly manipulation the spin states of electrocatalysts is seldom investigated, although this process generally demands tremendous energy and is kinetically sluggish, thus strongly retarding the transformation of NO₃⁻-to-NH₃.^{25,26}”

References 25 and 26 can not support the viewpoint “this process generally demands tremendous energy and is kinetically sluggish, thus strongly retarding the transformation of NO₃⁻-to-NH₃.” As

known, the kinetics of nitrite reduction and NO reduction are generally much faster than that of the NO₃⁻ reduction.

3. In this manuscript, the supporting data related to the catalyst performance is not adequately provided. At least, they should offer the standard curves for NH₃ quantification, the electrolysis curves at different applied potentials, the curves of cycling stability tests, and also the electrolysis conditions. These data can provide hints to identify whether the test is correct. Besides, the authors state that the catalyst shows 100% selectivity. How is the selectivity defined because they did not quantify other possible products?

4. In the main text, "A novel integrated equipment for flow-through NITRR electrolyzer and in-situ ammonia recovery".

What is the novelty of the flow-through electrolyzer, the structure, the working way, or the membrane? There is no information related to the flow-through electrolyzer and the testing procedures in the experimental section and the SI. What are the electrolyte volumes in the cathodic chamber or in the electrolyte container before and after the 200 hours of electrolysis?

Besides, as shown in Figure 3a, the current density is limited to ~75 mA cm⁻² at -0.4 V. Why does the current density reach 200 mA cm⁻² at -0.4 V in the flow-through electrolyzer? The authors ignored these important experimental details in the figure caption, in the main text, or in the SI, which make it impossible for others to discuss and compare the performance of the catalysts.

5. In the main text, the authors stated that "Compared with the Fe 2p spectra of SD-Fe₁-Ti, the Fe 2p 3/2 and Fe 2p 1/2 peaks of SP-Fe₁-Ti showed an obviously lower energy shift in binding energy, suggesting the reduction of Fe valence state induced by OVs and the appearance of electron-rich Fe 2+ species. As such, OVs on the amorphous TiO_x surface of SP-Fe₁-Ti can transfer electrons to the Fe single atoms and lead to an increased electron density of Fe single atoms."

In Figure 2b, the lower binding energy of Fe 2p peaks of SP-Fe₁-Ti compared to SD-Fe₁-Ti should be from the heat treatment of SP-Fe₁-Ti in H₂/Ar atmosphere and the treatment of SD-Fe₁-Ti in Ar/O₂ atmosphere, rather than the reduction of Fe valence state induced by OVs. As shown in Figure 2e, the SP-Fe₂-Ti and TiO_x/Ti show the same binding energy of Ti 2p peaks, suggesting that the loading Fe does not change the surface chemical state of Ti on the TiO_x/Ti with Ti³⁺ ion and OVs. Comparatively, the higher binding energy of the Ti 2p peak of SD-Fe₁-Ti should be due to the heat treatment under the Air/O₂ atmosphere, which results in the loss of Ti³⁺ and OVs, as indicated by Figure 1C. Thus, the author's conclusions are not valid.

6. In Figure 2f, the calculated μ_{eff} of SP-Fe1-Ti, SD-Fe1-Ti, and TiOx/Ti is 6.57, 3.46, and 1.20, respectively. The authors stated that the sample with a higher μ_{eff} value possesses much more spin electrons. According to the μ_{eff} values, the spin electrons of the catalysts should be mainly from the Fe rather than the TiOx/Ti with oxygen vacancies. Considering that the Fe in SP-Fe1-Ti is 2+, and the Fe in SD-Fe1-Ti is 3+, does it mean that the Fe²⁺ possesses more spin electrons than the Fe³⁺? So, it cannot be concluded that the presence of oxygen vacancies enhances the spin state of iron because the Fe valence state in SP-Fe1-Ti and SD-Fe1-Ti are different. There is insufficient experimental evidence to establish a direct correspondence between the spin state changes of Fe and the catalyst activity. Moreover, the TiOx/Ti substrate of SP-Fe1-Ti should have a higher conductivity than that of SD-Fe-Ti, due to its higher Ti³⁺ concentration, which may also impact the catalytic activities of Fe sites.

7. Why not provide the DEMS data of SD-Fe1-Ti? If SP-Fe1-Ti and SD-Fe1-Ti have differences in the *NO reduction step, there should be a difference in their DEMS spectra.

8. As shown in Figure 4e, the activation of Ti³⁺ caused by oxygen vacancies changes the bonding site from one to two, so how to define the number of oxygen vacancies in the DFT models? As known, it is easy to change the adsorption intensity of *NO or other intermediates by slightly changing the concentration of oxygen vacancies. Besides, the authors stated that the higher binding energy of -1.8 eV for NO on SP-Fe1-Ti than that of -1.58 eV on SD-Fe1-Ti suggests that the Fe1-Ti pairs were more favorable for the deoxygenation of NO₃⁻ to *NO. Why could the adsorption energy of *NO act as a descriptor for NO₃⁻ reduction activity?

Reviewer #3 (Remarks to the Author):

This study presents the remarkable performance of a spin-promoted Fe1-Ti pair on the oxygen vacancy-enriched TiOX (SP- Fe1-Ti) in the electrochemical nitrate reduction reaction (ENRR). In stark contrast to the spin-depressed Fe1-Ti pair on the oxygen vacancy-rare TiOX (SD- Fe1-Ti), the SP- Fe1-Ti catalyst demonstrated an impressive ammonia yield rate and nearly 100% selectivity for ammonia production across a wide range of potentials. Notably, high-quality NH₄Cl was successfully synthesized through continuous reaction. These findings highlight the significance of spin polarization in electrocatalysts for enhancing the performance of ENRR and provide valuable insights for the rational design of ENRR electrocatalysts. The manuscript exhibits a well-structured and well-written presentation of the research. Addressing the following concerns would further strengthen the paper for potential acceptance by Nature Communications.

1) My main concern is whether the discrepancies in the levels of oxygen vacancy and Ti³⁺ and the degree of crystallization also contribute to the differences in ENRR performance between SP-Fe1-Ti and SD-Fe1-Ti, in addition to the impact of spin polarization on the significantly improved performance of SP-Fe1-Ti.

2) The determination of the Fe dopant quantity in both SP-Fe1-Ti and SD-Fe1-Ti should be conducted using inductively coupled plasma (ICP) analysis. Additionally, it is recommended to compare the catalytic performance for ammonia yield based on mass activity, in addition to the electrochemical surface area (ECSA)-normalized activity. This comparison would provide a more comprehensive assessment of the catalytic performance of the materials.

3) Could you please provide information about the specific electrolyte used for NH₄Cl synthesis in the recovery chamber?

4) On page 12, it is mentioned that the recovery of ammonium via ENRR on SP-Fe1-Ti was tested at a current density of 200 mA cm⁻² for 200 h of electrolysis. There is a concern regarding whether the high current density and strong basic media could potentially compromise the hydrophobicity of the gas-permeable membrane. It is requested to provide the contact angle of the post-reacted membrane as a measure of its hydrophobicity.

5) In Figure 1c and Figure 2e, there are observed to be slight differences in the electron paramagnetic resonance (EPR) spectra and high-resolution Ti 2p X-ray photoelectron spectroscopy (XPS) spectra between TiOX/Ti and SP-Fe1-Ti. More explanations are suggested.

6) Figure 2e and Figure 2f exhibit a partial overlap in their data. It is recommended to revise the positioning of the data in these figures to ensure better visibility and clarity of the information presented.

7) In Figure 3b, the observed Faradaic efficiency (FE) being less than 100% suggests the occurrence of side reactions that contribute to the incomplete conversion of nitrate. Two potential side reactions that could affect the FE are the hydrogen evolution reaction (HER) and the nitrate-to-nitrite reaction. Please identify their contributions.

Point-by-point responses to the reviewers' comments

First of all, we thank the reviewers for their valuable comments and suggestions, which are very helpful to improve the quality and clarity of this paper. To address the specific concern/point clearly, we have separated the referees' comments into question areas and answered them in turn as follows.

Reviewer #1:

In this work, the authors successfully designed the spin-polarized Fe₁-Ti pairs by manipulating oxygen vacancies of the inherent surface oxide layer on monolithic Ti electrode for electrochemical nitrate reduction to ammonia (NITRR). The resultant electrode with spin-polarized Fe₁-Ti pairs could deliver extraordinary NITRR activity with an unprecedented NH₃ yield rate (272000 μg h⁻¹ mgcat⁻¹) and NH₃ Faradic efficiency (nearly 100 %) at -0.4 V vs. RHE, suppressing the counterpart of spin-depressed Fe₁-Ti electrode and most of the best-known electrodes. They clearly demonstrated that spin-polarized Fe-Ti pairs could provide spin electrons to interact with the reaction intermediates, thus facilitating the deoxygenation of NO₃⁻ to *NO and the hydrogenation of *NO via strengthening their adsorption. Moreover, the in-situ ammonia recovery with high recovery efficiency in this work further ensured the feasibility of the nitrate-to-ammonia conversion process from both energy and environment perspectives. This study provided a new diagram to develop the advanced electrodes with spin polarization effect towards spin-related reactions, which is very attractive for many applications. In addition, the analysis and experiments in this work are quite sound, and the manuscript is carefully organized and well written. We believe this work is definitely suitable for publication in Nature communications after some minor revisions. Some detailed comments are as below.

1. To further demonstrate the application prospect of the electrode with spin-polarized Fe₁-Ti pairs, the authors are suggested to evaluate the catalytic activity of the electrode at the neutral condition.

Response: We thank the reviewer for this valuable suggestion. As suggested by the reviewer, the catalytic activity of the spin-polarized Fe₁-Ti electrode at the neutral

condition was further evaluated. Impressively, the electrode with spin-polarized Fe₁-Ti pairs exhibited comparable NH₃ yield (15 mol h⁻¹ g_{cat}⁻¹) and FE (96.1%) to that of at the alkaline condition (**Figure R1**), demonstrating its high potential of environmental application.

During this revision, **Figure R1** was added into the revised Supporting Information as **Supplementary Figure 22** and the related discussion was added in the revised manuscript (**Line 276-280, Page 10 and 11**) as follows:

“We further evaluated the NITRR activity of the SP-Fe₁-Ti electrode at the neutral condition. Impressively, the SP-Fe₁-Ti exhibited comparable NH₃ yield rate (15 mol h⁻¹ g_{cat}⁻¹) and FE (96.1%) to that of at the alkaline condition (Supplementary Fig. 22), further demonstrating its high potential of environmental application.”

Figure R1. NH₃ yield rate and FE_{NH₃} of SP-Fe₁-Ti electrode at -0.4 V vs. RHE in 1 M KOH and 0.5 M Na₂SO₄ with addition of 1 M NaNO₃.

2. Regarding that the concentrations of NO₃⁻ in various waters are different, it is also necessary to check the impact of NO₃⁻ concentration on the electrode activity.

Response: We thank the reviewer very much for this valuable suggestion. As suggested by the reviewer, we measured the NITRR activity of SP-Fe₁-Ti electrode in the electrolyte with different NO₃⁻ concentrations (0.1, 0.5, and 1 M) at -0.4 V vs. RHE. As shown in **Figure R2**, the FEs of NO₃⁻-to-NH₃ conversion were 95.1%, 96.6% and 95.2% in the electrolyte with 0.1, 0.5, and 1 mol L⁻¹ NO₃⁻, respectively. These results suggest that the NO₃⁻ concentration has no obvious impacts on FE_{NH₃} of SP-Fe₁-Ti electrode, confirming the wide NO₃⁻ concentration compatibility. In addition, we observed that the NH₃ yield rate was enhanced by increasing the NO₃⁻ concentrations from 0.1 to 1 M because of the accelerated mass transfer (*Angew. Chem. Int. Ed.* 2022, 134,

During this revision, **Figure R2** was added into the revised Supporting Information as **Supplementary Figure 21** and the related discussion was added in the revised manuscript (**Line 268-276, Page 10**) as follows:

“Since the nitrate concentration varies in different sources, we also checked the NITRR activity of SP-Fe₁-Ti electrode in the electrolyte with different NO₃⁻ concentrations (0.1, 0.5, and 1 M) at -0.4 V vs. RHE. As shown in Supplementary Fig. 21, the FEs of NO₃⁻-to-NH₃ conversion were 95.1%, 96.6% and 95.2% in the electrolyte with 0.1, 0.5, and 1 M NO₃⁻, respectively. These results suggest that the NO₃⁻ concentration has no obvious impacts on FE_{NH₃} of SP-Fe₁-Ti electrode, indicating the wide NO₃⁻ concentration compatibility. In addition, we observed that the NH₃ yield rate was enhanced by increasing the NO₃⁻ concentrations from 0.1 to 1 M because of the accelerated mass transfer²³.”

Figure R2. NH₃ yield rate and FE_{NH₃} of SP-Fe₁-Ti electrode at -0.4 V vs. RHE in 1 M KOH with addition of 0.1, 0.5, 1 M NaNO₃.

3. It was widely reported that the hydrogen radical contributed to the nitrate reduction, so the authors should investigate the generation and the roles of hydrogen radicals to understand the mechanism of nitrate reduction with the electron spin resonance (ESR) technique.

Response: Thanks a lot for this valuable suggestion. During the revision, we employed the electron spin resonance (ESR) technique with 5,5-dimethyl-1-pyrroline-*N*-oxide (DMPO) as the radical trapping reagent to investigate the generation and the role of hydrogen radicals to understand the mechanism of nitrate reduction. As shown in

Figure R3, nine ESR peaks with an intensity ratio of 1:1:2:1:2:1:2:1:1 were observed for the electrocatalysis on the SP-Fe₁-Ti electrode in pure 1 M KOH, which could be assigned to the spin adduct of DMPO-H, confirming the generation of H_{ads}. When adding NaNO₃ into 1 M KOH, the signal intensity of DMPO-H apparently disappears. Such phenomenon revealed that the generated H_{ads} through water dissociation was consumed during the NITRR process, suggesting the deep participation of H* in the hydrogenation process of NITRR.

During this revision, **Figure R3** was added into the revised Supporting Information as **Supplementary Figure 30** and related discussion was added in the revised manuscript (**Line 314-323, Page 12**) as follows:

“We further employed the electron spin resonance (ESR) technique using 5,5-dimethyl-1-pyrroline-N-oxide (DMPO) as the radical trapping reagent to investigate the generation and the role of hydrogen radicals from water dissociation to understand the mechanism of nitrate reduction. Nine ESR peaks with an intensity ratio of 1:1:2:1:2:1:2:1:1 were observed in Supplementary Fig. 30 for the electrocatalysis on the SP-Fe₁-Ti electrode in pure 1 mol L⁻¹ KOH, which could be assigned to the spin adduct of DMPO-H, confirming the generation of H_{ads}. When adding NaNO₃ in 1 mol L⁻¹ KOH, the signal intensity of DMPO-H apparently disappeared. Such a phenomenon revealed that the generated H_{ads} through water dissociation was consumed, suggesting the deep participation of H in the hydrogenation process of NITRR.”*

Figure R3. ESR spectra of pristine electrolyte, the electrolyte obtained after 10 min electrocatalysis on SP-Fe₁-Ti electrode in 1 M KOH without NO₃⁻ and the electrolyte

obtained after 10 min electrocatalysis on SP-Fe₁-Ti electrode in 1 M KOH with NO₃⁻ under argon using DMPO as the ·H-trapping reagent.

4. Please provide the structural/electronic information of electrode and its Fe leaching concentration during the reaction to check the stability of electrode.

Response: We thank the reviewer very much for this kind comment. Further evidence to check the stability of SP-Fe₁-Ti electrode during NITRR is from XRD, XPS, and TEM analyses. We did not observe any change of XRD and XPS peaks as well as TEM images of SP-Fe₁-Ti electrode after reaction (**Figure R4-6**), suggesting its high stability under NITRR conditions. Moreover, we also monitored the dissolved Fe ions of SP-Fe₁-Ti electrode during NITRR in electrolyte to investigate the stability of electrode. Only a negligible amount of Fe ions (**Figure R7**) was detected after 20 cycles consecutive electrolysis, again confirming the excellent stability of SP-Fe₁-Ti electrode.

During this revision, **Figure R4-7** were added into the revised Supporting Information as **Supplementary Figure 24-27** and the related discussion was added in the revised manuscript (**Line 284-287, Page 11**) as follows:

“XRD, XPS, and TEM analyses further revealed the high stability of SP-Fe₁-Ti electrode during NITRR (Supplementary Figs. 24-26), as confirmed by the negligible Fe dissolution of SP-Fe₁-Ti electrode within 20 cycles consecutive electrolysis (Supplementary Fig. 27).”

Figure R4. XRD patterns of pristine and post-reacted SP-Fe₁-Ti electrode.

Figure R5. (a) HRTEM image and (b) HADDF-STEM image and corresponding elemental mapping of post-reacted SP-Fe₁-Ti electrode.

Figure R6. (a) Fe 2p XPS spectra of pristine and post-reacted SP-Fe₁-Ti electrode. (b) Ti 2p XPS spectra of pristine and post-reacted SP-Fe₁-Ti electrode.

Figure R7. Fe ions concentrations of the pristine electrolyte and the post-reacted electrolyte after different cycles consecutive electrolysis.

5. ¹H NMR spectra of the original electrolyte containing ¹⁵NO₃⁻ are required to confirm that the produced NH₃ was originated from the nitrate feedstock instead of contaminations.

Response: We thank the reviewer very much for this valuable suggestion. Peaks of $^{15}\text{NH}_4^+$ were absent in ^1H NMR spectra of the original electrolyte containing $^{15}\text{NO}_3^-$ (Figure R8), confirming that the produced NH_3 was originated from the reduction of nitrate feedstock instead of contaminations.

During this revision, ^1H NMR spectra of the original electrolyte containing $^{15}\text{NO}_3^-$ was added in Figure 3d and the related discussion was added in the revised manuscript (Line 255-258, Page 10) as follows:

“In addition, peaks of $^{15}\text{NH}_4^+$ are absent in ^1H NMR spectra of the original electrolyte containing $^{15}\text{NO}_3^-$, further confirming that the produced NH_3 was originated from the reduction of nitrate feedstock instead of contaminations.”

Figure R8. ^1H NMR spectrum of the generated products during the electrocatalytic NITRR using $^{14}\text{NO}_3^-$ and $^{15}\text{NO}_3^-$ as reagents on the SP- Fe_1 -Ti electrode at -0.4 V vs. RHE and original electrolyte containing $^{15}\text{NO}_3^-$.

Reviewer #2:

This paper reported a catalyst based on Fe single atom-modified TiO_x/Ti substrate for NO_3^- -to- NH_3 conversion (NITRR). They suggested that the oxygen vacancies in the TiO_x/Ti substrate could activate the Fe sites and produce spin-polarized Fe_1 -Ti pairs for enhanced NITRR performance, compared to the counterpart that underwent heat treatment under the Air/ O_2 atmosphere. Nevertheless, the activity of the catalysts did not reach the “superior” state they claimed, and there is insufficient experimental

evidence to establish a direct correspondence between spin state changes and activity. More detailed comments are below.

1. In this paper, the authors stated that “The designed spin-polarized Fe₁-Ti pairs delivered an unprecedented NH₃ Faradic efficiency of nearly 100% and an ultrahigh NH₃ yield rate of 272000 μg h⁻¹ mg_{cat}⁻¹ at -0.4 V vs. RHE for NITRR, far superior to the counterpart monolithic electrode with spin-depressed Fe₁-Ti pairs (51000 μg h⁻¹ mg_{cat}⁻¹) and one order of magnitude higher than mostly reported NITRR electrocatalysts.”.

Now, numerous catalysts achieve a nearly 100% NH₃ Faradic efficiency even at more positive applied potentials (>-0.2 V vs. RHE). Besides, It is not reasonable to define the NH₃ yield rate by the mass of the Fe single atoms in this catalyst because 1) the Ti sites might take part in the catalytic progress as suggested by the DFT calculations; 2) the price of Ti substrate is much higher than that of the Fe. On the other hand, the performance of the SP-Fe₁-Ti is only average, as evidenced by the negative onset potentials (~-0.1 to -0.2 V) and a low current density of ~75 mA cm⁻² in electrolytes even with 1 M NO₃⁻ at -0.4 V. As known, the reported catalysts related to the CuRu, CoRu, CuCo show a much better performance than this SP-Fe₁-Ti catalyst.

For a further efficient comparison with other newly reported catalysts, it is necessary to provide the NH₃ yield rate defined by the electrode area, or the sum mass of the electrode (or the high-price components) (the ways used in industry). Note that the progress of an electrocatalytic topic requires a reasonable way to define the real performance of the reported catalysts. It doesn't make sense to get an "eye-catching" number in a biased way.

Response: We thank the reviewer very much for the valuable comments and suggestions. First, we thoroughly reviewed literatures and compared the applied potentials for maximizing FE_{NH3}, onset potentials and current densities of top-level NITRR electrocatalysts. As shown in **Figure R9** and **Table R1**, the applied potential for maximizing FE_{NH3} and onset potential of SP-Fe₁-Ti electrode were lower than almost all of top-level NITRR electrodes consisting of the low-price components (e.g., Cu, Ti, and Fe). More importantly, SP-Fe₁-Ti electrode exhibited a higher current density at

-0.4 V vs. RHE than almost all of top-level NITRR electrodes consisting of the low-price components (e.g., Cu, Ti, and Fe). Only few electrodes containing Co metal or even noble Ru and Pd metals exhibited a better performance. However, the price of Co, Ru or Pd metal is much higher than that of Ti and Fe (**Table R2**).

We do agree that the Reviewer #2's suggestion on providing a reasonable way to define the real performance of the reported catalysts, which could help us to efficiently and fairly compare their performance. During the revision, we calculated the NH₃ yield rate defined by the electrode area for performance comparison. It is worth noting that almost all of the top-level NITRR electrocatalysts in previous studies were in form of powders. Commonly, they were drop-casted on the carbon-based supports for performance assessment. Likewise, we crumbled the monolithic electrode into catalyst powders, drop-casted the catalyst-containing ink onto the carbon fiber paper with the catalyst mass loading of 1 mg cm⁻² and then evaluated its NITRR performance. As shown in **Figure R10**, SP-Fe₁-Ti catalyst displays a high cathodic current density of 190 mA cm⁻² at -0.4 V vs. RHE, superior to most of the top-level NITRR catalysts. The NH₃ yield rate defined by the electrode area of SP-Fe₁-Ti catalyst was determined to be 0.99 mmol cm⁻² h⁻¹, which outperforms most of low-price metal-based catalysts and even several high-price metal-based catalysts (**Table R3**). More importantly, the monolithic nature of SP-Fe₁-Ti electrode is much more feasible than the powder form of previous top-level NITRR electrocatalysts for practical application.

To avoid misunderstanding, the title of "Spin polarized Fe₁-Ti pairs for superior electroreduction nitrate to ammonia" was revised to "Spin polarized Fe₁-Ti pairs for highly efficient electroreduction nitrate to ammonia". Additionally, **Figure R10** and **Table R1-3** were added into the revised Supporting Information as **Supplementary Figure 16** and **Supplementary Table 4-6**, and the related discussion was added in the revised manuscript (**Line 229-242, Page 9**) as follows:

"Additionally, SP-Fe₁-Ti electrode possesses promising application potential for maximizing FE_{NH_3} , onset potential and current density at certain potential (Supplementary Table 4) over almost all of top-level NITRR electrodes consisting of the low-price components (e.g., Cu, Ti, and Fe in Supplementary Table 5). We also drop-casted SP-Fe₁-Ti catalyst powders on the carbon fiber paper with the catalyst

mass loading of 1 mg cm^{-2} for performance assessment. As shown in Supplementary Fig. 16, SP-Fe₁-Ti catalyst displays a high cathodic current density of 190 mA cm^{-2} at -0.4 V vs. RHE , superior to most of the top-level NITRR catalysts. The NH₃ yield rate defined by the electrode area of SP-Fe₁-Ti catalyst was determined to be $0.99 \text{ mmol cm}^{-2} \text{ h}^{-1}$, which outperforms most of low-price metal-based catalysts and even several high-price metal-based catalysts (Supplementary Table 6). More importantly, the monolithic nature of SP-Fe₁-Ti electrode is much more feasible than the powder form of previous top-level NITRR electrocatalysts for practical application.”

Figure R9. Comparison of the applied potential for maximizing FE_{NH_3} of top-level NITRR electrodes consisting of the low-price components (e.g., Cu, Ti, and Fe).

Figure R10. LSV curves of SP-Fe₁-Ti powders coated onto the carbon fiber paper with and without addition of NO_3^- .

Table R1. Comparison of the applied potentials for maximizing FE_{NH_3} , onset potentials and current density of top-level NITRR electrocatalysts.

Catalysts	Applied potentials for maximizing	Onset potential (V)	Current density @ -0.4 V vs. RHE	References
-----------	-----------------------------------	---------------------	--	------------

	E_{NH_3} (V vs. RHE)	vs. RHE)	(mA cm ⁻²)	
			~75 (monolithic electrode)	
SP-Fe ₁ -Ti	-0.4	~-0.1	~190 (powder)	This work
Fe single atom	-0.66	~-0.3	~5	Nat. Commun., 2021, 12, 2870.
Fe-PPy SACs	-0.3	~0.3	~12	Energy Environ. Sci., 2021,14, 3522
FeB	-0.6	~-0.2	~150	Angew. Chem. Int. Ed., 2023, e202300054
Fe ₂ TiO ₅	-0.9	~-0.1	~18	Angew. Chem. Int. Ed., 2023, 62, e202215782
SA-Fe(II)	-1.0	~-0.3	~10	Proc. Natl. Acad. Sci., 2023, 120, e2209979120.
CuCl/TiO ₂	-0.8	~-0.3	~8	Angew. Chem. Int. Ed. 2021, 60, 22933.
Cu/Cu ₂ O NWAs	-0.85	~-0.2	~35	Angew. Chem. Int. Ed.2020,59, 5350–5354
Cu-PTCDA	-0.4	0.27	~15	Nat. Energy., 2020, 5, 605.
Co-Fe@Fe ₂ O ₃	-0.75	~-0.2	~5	Proc. Natl. Acad. Sci., 2022, 119, e2115504119.
CoP-CNS	-1.03	~0	~200	Nat. Commun., 2022, 13, 7958.
CoP NAs	-0.3	~0.1	~300	Energy Environ. Sci., 2022,15, 760.
Ru ₁ -TiO _x /Ti	-0.3	~0.1	~100	Angew. Chem. Int. Ed. 2022, 61,

				e202208215
Strained Ru	-0.2	~0.2	~110	J. Am. Chem. Soc., 2020, 142, 7036.
Ru ₁₅ Co ₈₅ HNDs	0	~0.4	/	Nat Catal 2023, 6, 402.
CuCo nanosheet	-0.2 V	~0.1	/	Nat Commun 2022, 13, 7899
Ru-CuNW	0.04	~0.2	/	Nat. Nanotechnol. 2022, 17, 759.
Ru ₁ Cu ₁₀ /rGO	-0.05	~0.4	/	Adv. Mater. 2023, 35, 2202952.
CuPd	-0.6	~0	~180	Nat. Commun., 2022, 13, 2338.

Table R2. Price of different metals^[a]

Metal	Symbol	Unit of Measure	Price
Palladium	Pd	g	86.72 \$
Ruthenium	Ru	g	24.11 \$
Cobalt	Co	g	0.033 \$
Copper	Cu	g	0.0082 \$
Titanium	Ti	g	0.0063 \$
Iron	Fe	g	0.00010 \$

[a] The prices for various metals are from the metalary & tradingeconomics website on July 13, 2023. (<https://www.metalary.com>; <https://tradingeconomics.com>)

Table R3. Comparison of the NH₃ yield rate defined by the electrode area among various electrodes.

Catalysts	FE _{NH3}	NH ₃ yield rate (mmol cm ⁻² h ⁻¹)	References
SP-Fe ₁ -Ti	98.51% at -0.4 V vs. RHE	0.99	This work
Fe single atom	~ 75% at -0.66 V vs. RHE	0.12	Nat. Commun., 2021, 12, 2870.
Fe-PPy SACs	99.69% at -0.3 V vs. RHE	0.16	Energy Environ. Sci., 2021,14, 3522

FeB	96.8 % at -0.6 V vs. RHE	1.5	Angew. Chem. Int. Ed., 2023, e202300054
Fe ₂ TiO ₅	87.6 % at -0.9 V vs. RHE	0.073	Angew. Chem. Int. Ed., 2023, 62, e202215782
SA-Fe(II)	99.6 % at -1.0 V vs. RHE	0.29	Proc. Natl. Acad. Sci., 2023, 120, e2209979120.
CuCl/TiO ₂	85 % at -0.8 V vs. RHE	0.13	Angew. Chem. Int. Ed. 2021, 60, 22933.
Cu/Cu ₂ O NWAs	95.8 % at -0.85 V vs. RHE	0.24	Angew. Chem. Int. Ed.2020,59, 5350 –5354
Cu-PTCDA	77 % at -0.4 V vs. RHE	0.026	Nat. Energy., 2020, 5, 605.
Co-Fe@Fe ₂ O ₃	85.2 % at -0.75 V vs. RHE	0.089	Proc. Natl. Acad. Sci., 2022, 119, e2115504119.
CoP-CNS	88.6 % at -1.03 V vs. RHE	8.47	Nat. Commun., 2022, 13, 7958.
CoP NAs	~100 % at -0.3 V vs. RHE	3.09	Energy Environ. Sci., 2022,15, 760.
Ru ₁ -TiO _x /Ti	87.3 % at -0.3V vs. RHE	/	Angew. Chem. Int. Ed. 2022, 61, e202208215
Strained Ru	~100 % at -0.2 V vs. RHE	1.03	J. Am. Chem. Soc., 2020, 142, 7036.
Ru ₁₅ Co ₈₅ HNDs	97 % at 0 V vs. RHE	1.92	Nat Catal 6, 402–414 (2023).
CuCo nanosheet	100 % at -0.2 V vs. RHE	4.8	Nat Commun 13, 7899 (2022)
Ru-CuNW	96 % at 0.04 V vs. RHE	4.5	Nat. Nanotechnol. 17, 759–767 (2022).
Ru ₁ Cu ₁₀ /rGO	98 % at -0.05 V vs. RHE	0.38	Adv. Mater. 2023, 35, 2202952.
CuPd	92.5 % at -0.6 V vs. RHE	1.25	Nat. Commun., 2022, 13, 2338.

2. In the introduction, “While the acceleration of spin–transition related *NO hydrogenation via properly manipulation the spin states of electrocatalysts is seldom investigated, although this process generally demands tremendous energy and is kinetically sluggish, thus strongly retarding the transformation of NO₃[–] to NH₃.^{25,26}” References 25 and 26 cannot support the viewpoint “this process generally demands tremendous energy and is kinetically sluggish, thus strongly retarding the transformation of NO₃[–] to NH₃.” As known, the kinetics of nitrite reduction and NO reduction are generally much faster than that of the NO₃[–] reduction.

Response: Thanks a lot for these valuable comments. In reference 26, the authors summarized that the change in the electronic spin multiplicity state between reactants and products would be the origin of the high overpotential for slow reaction kinetics of the spin-related reactions (e.g., OER and ORR) as follows:

“In chemistry, electron spin is a primary factor that controls chemical reactions, besides the free and activation energies. Conservation of spin angular momentum is generally a fundamental property of nature besides the energy conservation, and it forms the basis of the so-called spin selection rule (Wigner’s spin conservation rule), which states that in any allowed electronic transition process, either interatomically or intra-atomically, the spin angular momentum of the system should not change, and the transition between two states of different spin multiplicity is forbidden. Applying this to chemistry, a chemical reaction is allowed only if the total spin angular momentum of reactants is same as the total spin angular momentum manifested by the products, resulting in electron and nuclear spin selectivity of reactions. For instance, the combination of two radicals with the unpaired electrons aligned parallelly toward the formation of net zero spin species is strictly forbidden.”

Reference 25 also agreed with such an opinion and the authors concluded that “the overpotential required to split water is linked to restrictions on the electrons’ spin in generating a ground state triplet oxygen molecule”. They found that “During water splitting, two OH[–] species must combine to form molecular oxygen in its triplet ground state. In the process, an electron from each OH[–] is transferred to the anode. This leaves the two OH• radicals in their doublet ground state, namely each OH• has one unpaired electron. When there is no spin control and the interaction electronic potential has a

singlet character, the formation of H₂O₂ is possible. However, when the electron's spins are aligned in a parallel fashion, the two electrons interact on the triplet potential surface which correlates with the formation of the ground state molecular oxygen and on which the formation of H₂O₂ is symmetry forbidden.” and concluded that “controlling the spin state of the electronic potential on which the reaction occurs should result in more efficient oxygen production and limited production of hydrogen peroxide.”

Therefore, references 25 and 26 clearly conveyed the viewpoint that *“the spin-related process generally demands tremendous energy and is kinetically sluggish, thus strongly retarding the transformation process.”*

As stated in the introduction of our manuscript (Line 63 and 64, Page 3), *“*NO hydrogenates to the hydrogenated *NHO/*NOH species with a spin state transition”,* which was similar to the spin-forbidden OER/ORR process. Moreover, the conversion of *NO to *NHO/*NOH during NITRR has been intensively reported to be the rate-determining step and strongly retards the transformation of NO₃⁻ to NH₃ in many reported studies (Nat. Commun., 2021, 12, 2870; Energy Environ. Sci., 2021, 14, 3522; Angew. Chem. Int. Ed. 2021, 60, 22933; Angew. Chem. Int. Ed. 2020, 59, 5350; Nat. Commun., 2022, 13, 7958), but the acceleration of spin-transition related *NO hydrogenation to *NHO/*NOH via properly manipulating the spin states of electrocatalysts is seldom investigated.

To avoid the misunderstanding, we revised *“While the acceleration of spin-transition related *NO hydrogenation via properly manipulation the spin states of electrocatalysts is seldom investigated, although this process generally demands tremendous energy and is kinetically sluggish, thus strongly retarding the transformation of NO₃⁻ to NH₃^{25,26.}”* to *“While the acceleration of spin-transition related *NO hydrogenation via properly manipulation the spin states of electrocatalysts is seldom investigated, although this process is similar with reported oxygen-related electrocatalytic reactions such as oxygen evolution reaction (OER) and oxygen reduction reaction (ORR) processes^{25,26,} and generally demands tremendous energy and is kinetically sluggish, thus strongly retarding the transformation of NO₃⁻ to NH₃.”*

3. In this manuscript, the supporting data related to the catalyst performance is not adequately provided. At least, they should offer the standard curves for NH_3 quantification, the electrolysis curves at different applied potentials, the curves of cycling stability tests, and also the electrolysis conditions. These data can provide hints to identify whether the test is correct. Besides, the authors state that the catalyst shows 100% selectivity. How is the selectivity defined because they did not quantify other possible products?

Response: We thank the reviewer very much for the valuable comments and suggestions. During the revision, we added the standard curves for NH_3 quantification (**Figure R11**), the electrolysis curves at different applied potentials (**Figure R12**), and the curves of cycling stability tests (**Figure R13**) in the revised Supplementary Information. Actually, the electrolysis conditions were provided in our original manuscript (**Electrochemical measurements in Method**). For the sake of convenience, the related information was also supplemented as below:

“NITRR test was conducted in a typical H-type cell, separated by a Nafion 117 membrane. The $\text{Fe}_1\text{-Ti}$ electrodes with specific spin polarization degree ($\text{SP-Fe}_1\text{-Ti}$ and $\text{SD-Fe}_1\text{-Ti}$), platinum plate and Hg/HgO electrode were used as working electrode, counter electrode and reference electrode, respectively. If not specified otherwise, all potentials reported in this work were referenced to a reversible hydrogen electrode (RHE). Before each measurement, the electrolyte was purged with Ar (99.99%) for 30 min. Potentiostatic tests were conducted at certain applied potentials within NITRR window for 2 h.”

To check the contributions of other possible products (e.g., H_2 or NO_2^-) to the faradaic current, we investigated possible gas products using gas chromatography (GC, Thermal Trace-1300) equipped with thermal conductivity detector (TCD) and possible liquid products using UV-Vis spectrometry. As shown in **Figure R14**, the only detected gas product was H_2 , and the only detected liquid product was NO_2^- . The Griess test was used to determine the NO_2^- concentration in electrolytes after the potentiostatic test. Generally, N-1-naphthyl ethylenediamine dihydrochloride (0.04 g), p-aminobenzene sulfonamide (0.8 g), and H_3PO_4 (2 mL, 85%) were dissolved in DI

water (10 mL) as the Griess reagent. The diluted electrolyte (5 mL) was mixed with the Griess agent (100 μ L) and rested for 10 min at room temperature to conduct UV-vis tests. NO_2^- concentrations of the electrolyte were determined by the absorbance at 540 nm. In the same operation, various NaNO_2 aqueous solutions were used as the standard samples to obtain the calibration curve (**Figure R15**).

The FE of product i (FE_i) was calculated via the following relation:

$$\text{FE}_i = (n_i \times F \times c_i \times V) / (17 \times Q)$$

where n_i is the number of electron transferred to product i , c_i is the measured product i concentration, V is the volume of the electrolyte or the upper space, F is the Faraday constant (96485 C mol^{-1}) and Q is the total charge.

As shown in **Figure R16**, the FEs of H_2 and NO_2^- were calculated to be 4.1% and 0.7%, respectively, at -0.4 V vs. RHE. After taking the two side-reactions into consideration, the total FE was 100% as expected. We are sorry not to calculate the selectivity of NH_3 in our original manuscript. As suggested by Reviewer #2, the selectivity (S) of NH_3 was determined to be 95.9% at -0.4 V vs. RHE according to the equation:

$$S_{\text{NH}_3} = C_{\text{NH}_3} / (C_{\text{NH}_3} + C_{\text{NO}_2^-})$$

During this revision, **Figure R11-16** were added into the revised Supporting Information as **Supplementary Figure 13, 14 and 17-19** and related discussion was added in the revised manuscript (**Line 243-250, Page 9 and 10**) as follows:

“The contributions of other possible products to the faradaic current were also evaluated using gas chromatography equipped with thermal conductivity detector (TCD) and UV-Vis spectrometry. Only H_2 and NO_2^- were detected (Supplementary Fig. 17 and Supplementary Fig. 18), and their FEs were calculated to be 0.7% and 4.1%, respectively, at -0.4 V vs. RHE. After taking the two side-reactions into consideration, the total FE (Supplementary Fig. 19) was 100% as expected. Moreover, the NH_3 selectivity of SP-Fe₁-Ti electrode was determined to be 95.9% at -0.4 V vs. RHE, further confirming its high selectivity towards NH_3 production from NITRR.”

Figure R11. Standard calibration curves for UV-Vis detection of NH_3 from the Nessler's method (a) the standard solutions (b) raw UV-Vis spectra (b) linear calibration.

Figure R12. The electrolysis curves of (a) $\text{SP-Fe}_1\text{-Ti}$ and (b) $\text{SD-Fe}_1\text{-Ti}$ at different applied potentials.

Figure R13. The curves of cycling stability tests.

Figure R14. Representative GC of gas products obtained on SP-Fe₁-Ti electrode at -0.4 V vs. RHE.

Figure R15. Standard calibration curves for UV-Vis detection of NO₂⁻ from the Griess's method (a) the standard solutions (b) raw UV-Vis spectra (c) linear calibration.

Figure R16. The FE of H₂ and NO₂⁻ and NH₃ during NITRR on on SP-Fe₁-Ti electrode at -0.4 V vs. RHE.

4. In the main text, “A novel integrated equipment for flow-through NITRR electrolyzer and in-situ ammonia recovery”. What is the novelty of the flow-through electrolyzer, the structure, the working way, or the membrane? There is no information related to the flow-through electrolyzer and the testing procedures in the experimental section and the SI. What are the electrolyte volumes in the cathodic chamber or in the electrolyte container before and after the 200 hours of electrolysis? Besides, as shown in Figure 3a, the current density is limited to $\sim 75 \text{ mA cm}^{-2}$ at -0.4 V . Why does the current density reach 200 mA cm^{-2} at -0.4 V in the flow-through electrolyzer? The authors ignored these important experimental details in the figure caption, in the main text, or in the SI, which make it impossible for others to discuss and compare the performance of the catalysts.

Response: We thank the reviewer very much for the valuable comments and suggestions. In this work, we designed an integrated device composed of a flow-through NITRR electrolyzer and a membrane-based ammonia recovery unit to simultaneously realize the efficient NITRR and in-situ NH_3 recovery. Such a novel device could enable the continuous production of NH_3 from NO_3^- reduction, and also achieve the separation of NH_3 product from the electrolyte without additional energy input, which is critical for practical downstream applications.

We are sorry for unclear statement regarding the selection of anode to evaluate the NITRR performance. In this study, $1 \times 1 \text{ cm}$ Pt plate was used as the anode to evaluate the NITRR performance of SP- Fe_1 -Ti electrode in a H-cell, while the commercial dimension stable anode (DSA) with a size of $2 \times 2 \text{ cm}$ was used in the flow-through electrolyzer because large-scale Pt plate is expensive and OER-inactive compared to DSA, as evidenced by the LSV curves in **Figure R17**. Moreover, the flow-through mode could significantly accelerate the mass transfer and reduce ohmic losses of reactions. Therefore, the larger electrode size, better OER activity and the accelerated mass transfer in a flow-through way collectively resulted in a higher current density compared to that of in a H-type cell.

In this work, 250 mL 1 M NO_3^- -containing solution was adopted to assess the efficiency of the flow-through coupled device. During operation, the NO_3^- -containing solution flowed into the electrocatalytic cell to be reduced into NH_3 -containing

electrolyte, which was directly guided into an acidic adsorption solution (1 M HCl solution, 800 mL) via a hollow PP fiber membrane, only allowing the gaseous NH_3 to diffuse across the membrane for its recovery as high-purity ammonium. To keep the efficient production of NH_3 , we replaced the electrolyte in the cathodic chamber to maintain the NO_3^- concentration every day.

For better understanding, more detailed information regarding the operation conditions of the integrated flow-through coupled device was added in the revised manuscript (**Line 406-413, Page 15**) as follows:

“We designed a compact flow-through electrolyzer consisting of SP-Fe₁-Ti cathode and commercial dimension stable anode (DSA) as the NITRR electrolyzer. During operation, the NO_3^- -containing solution (250 mL) as synthetic effluent was treated through electrochemical reduction on SP-Fe₁-Ti cathode, and circulated by a peristaltic pump with a flow rate of 60 mL min^{-1} . Then the NH_3 -containing effluent was directly guided into the hollow polypropylene (PP) fiber arrays immersed into 1 M HCl solution (800 mL) to recover NH_3 .”

Figure R17. LSV curve of the flow-through electrolyzer with the compact structure and the employment of the OER-active DSA electrode

5. In the main text, the authors stated that “Compared with the Fe 2p spectra of SD-Fe₁-Ti, the Fe 2p 3/2 and Fe 2p 1/2 peaks of SP-Fe₁-Ti showed an obviously lower energy shift in binding energy, suggesting the reduction of Fe valence state induced by OVs and the appearance of electron-rich Fe 2+ species. As such, OVs on the amorphous TiO_x surface of SP-Fe₁-Ti can transfer electrons to the Fe single atoms and

lead to an increased electron density of Fe single atoms.”

In Figure 2b, the lower binding energy of Fe 2p peaks of SP-Fe1-Ti compared to SD-Fe1-Ti should be from the heat treatment of SP-Fe1-Ti in H₂/Ar atmosphere and the treatment of SD-Fe1-Ti in Ar/O₂ atmosphere, rather than the reduction of Fe valence state induced by OVs. As shown in Figure 2e, the SP-Fe1-Ti and TiO_x/Ti show the same binding energy of Ti 2p peaks, suggesting that the loading Fe does not change the surface chemical state of Ti on the TiO_x/Ti with Ti³⁺ ion and OVs. Comparatively, the higher binding energy of the Ti 2p peak of SD-Fe1-Ti should be due to the heat treatment under the Air/O₂ atmosphere, which results in the loss of Ti³⁺ and OVs, as indicated by Figure 1C. Thus, the author’s conclusions are not valid.

Response: Thanks a lot for these valuable comments. We do agree the reviewer’s constructive opinion that the thermal treatment under different atmosphere plays a vital role in manipulating the electronic properties of Fe and Ti in this study. Meanwhile, we are sorry for unclear discussion on the interaction between oxygen vacancies (OVs) and Fe or Ti. During hydrogen treatment of transition metal oxides (TMOs), it is widely accepted that electrons are first transferred from hydrogen (H) atoms to the oxygen (O) atoms in the lattice of TMOs. Then, the lattice O leaves with the H atom to form H₂O, and the oxygen vacancies form on the surface of TMOs. Finally, the electrons located on the oxygen vacancy states are transferred to neighboring TMⁿ⁺ and thus reduced TMⁿ⁺ to TM⁽ⁿ⁻¹⁾⁺ (Nature, 2004, 430, 657; Nat. Mater., 2014, 13, 488; Phys. Chem. Chem. Phys., 2013, 15, 2117; J. Phys. Condens. Matter, 2013, 25, 236002). In our case, we detected an obvious H₂O evolution peak at about 370 °C during the temperature programmed reaction (TPR) measurement of pristine Ti foam in 5% H₂/Ar (**Figure R18**), consequently generating OVs on the TiO_x/Ti foam, as evidenced in the electron spin resonance (ESR) spectra (**Figure R19**). The unpaired electrons located in OVs can be transferred to the empty 3d levels at the bottom of the conduction band belonging to the adjacent Ti atoms. Since the 3d states are rather localized, this electron transfer corresponds to a change of the formal oxidation state from Ti⁴⁺ to Ti³⁺, resulting in two observable consequences such as a negative shift in the core level binding energies of the reduced Ti atoms (**Figure R20**), and the presence of an unpaired electron (spin) in the 3d shell of the Ti atoms (**Figure R19**). Similarly, Fe³⁺

anchored on OV_s can be reduced to Fe²⁺ by unpaired electrons located on OV_s, as evidenced by the emerge of Fe²⁺ species in the Fe 2p XPS spectrum (**Figure R21**) of FeCl₃/TiO_x even without thermal treatment. As mentioned by Reviewer #2, there was no electronic interaction between Fe and Ti, which was confirmed by the unchanged binding energy of Ti 2p peaks after Fe loading in **Figure R20**. Therefore, OV_s with the unpaired electrons formed in hydrogen treatment, are the prerequisite for the reduction of valence state for Fe and Ti in our study. For this reason, the electronic properties of Fe and Ti in SD-Fe₁-Ti electrode without OV_s were different from those of SP-Fe₁-Ti electrode, maintaining Fe³⁺ and Ti⁴⁺ states.

For better understanding, **Figure R18 and R21** were added into the revised Supporting Information as **Supplementary Figure 3 and 12** and the related discussion was added in the revised manuscript (**Line 123-128, Page 5; Line 161-167, Page 6 and 7; Line 183-185, Page 7**) as follows:

Line 123-128, Page 5: *“During hydrogen treatment of inherent oxide layer of Ti foam, electrons were first transferred from hydrogen (H) atoms to the oxygen (O) atoms in the lattice of inherent oxide layer. Then, the lattice O leaves with the H atom to form H₂O, as evidenced by an obvious H₂O evolution peak at about 370 °C during the temperature programmed reaction (TPR) measurement of pristine Ti foam in 5% H₂/Ar (Supplementary Fig. 3), and the OV_s form on the surface of TiO_x/Ti foam.”*

Line 161-167, Page 6 and 7: *“The unpaired electrons located in OV_s are expected to transfer to the empty 3d levels belonging to Ti and Fe atoms adjacent to OV_s^{36, 37}, resulting in two possible consequences including a negative shift in the core level binding energies of the reduced Ti or Fe atoms, and the presence of an unpaired electron (spin) in the 3d shell of the Ti or Fe atoms^{38, 39}. While the electronic properties of Fe and Ti in SD-Fe₁-Ti electrode without OV_s should be different from those of SP-Fe₁-Ti electrode, maintaining spin-depressed Fe³⁺ and Ti⁴⁺ states.”*

Line 183-185, Page 7: *“Such an electron transfer from OV_s to Fe atoms even occurs within FeCl₃/TiO_x without thermal treatment, as evidenced by the appearance of Fe²⁺ species in its Fe 2p XPS spectra (Supplementary Fig. 12).”*

Figure R18. Mass spectroscopy during TPR of Ti foam.

Figure R19. ESR spectra of TiO_x/Ti .

Figure R20. Ti 2p XPS spectra of Ti foam, TiO_x/Ti and $\text{SP-Fe}_1\text{-Ti}$.

Figure R21. Fe 2p XPS spectra of $\text{FeCl}_3/\text{TiO}_x$ without thermal treatment, $\text{SD-Fe}_1\text{-Ti}$, and $\text{SP-Fe}_1\text{-Ti}$.

6. In Figure 2f, the calculated μ_{eff} of $\text{SP-Fe}_1\text{-Ti}$, $\text{SD-Fe}_1\text{-Ti}$, and TiO_x/Ti is 6.57, 3.46, and 1.20, respectively. The authors stated that the sample with a higher μ_{eff} value possesses much more spin electrons. According to the μ_{eff} values, the spin electrons of the catalysts should be mainly from the Fe rather than the TiO_x/Ti with oxygen vacancies. Considering that the Fe in $\text{SP-Fe}_1\text{-Ti}$ is 2+, and the Fe in $\text{SD-Fe}_1\text{-Ti}$ is 3+, does it mean that the Fe^{2+} possesses more spin electrons than the Fe^{3+} ? So, it cannot be concluded that the presence of oxygen vacancies enhances the spin state of iron because the Fe valence state in $\text{SP-Fe}_1\text{-Ti}$ and $\text{SD-Fe}_1\text{-Ti}$ are different. There is insufficient experimental evidence to establish a direct correspondence between the spin state changes of Fe and the catalyst activity. Moreover, the TiO_x/Ti substrate of $\text{SP-Fe}_1\text{-Ti}$ should have a higher conductivity than that of $\text{SD-Fe}_1\text{-Ti}$, due to its higher Ti^{3+} concentration, which may also impact the catalytic activities of Fe sites.

Response: We thank the reviewer very much for these constructive comments. First, we would like to re-emphasize that the spin electrons of $\text{SP-Fe}_1\text{-Ti}$ are from both the spin-polarized Fe and spin-polarized Ti, but the spin electrons of the $\text{SD-Fe}_1\text{-Ti}$ are only from Fe in view of Ti^{4+} without any 3d electrons. Meanwhile, experimental and theoretical results have demonstrated the contribution of the spin-polarized Ti^{3+} towards the electrode's activity (**Figure 3f, 4h, 4i and Supplementary Fig. 33**). So, it is not reasonable to establish a direct relationship between the spin state change of Fe and the catalyst activity. Therefore, the spin polarization of $\text{Fe}_1\text{-Ti}$ pairs was selected

as the activity descriptor to clarify their activity differences in this work.

Moreover, it should be noted that spin-state refers to the potential spin configurations of the transition metal d electron. Based on crystal field theory, transition metal possesses multiple spin-states with different orbital electronic configurations. For example, Fe³⁺ is defined as low spin (LS), intermediate-spin (IS) and high-spin (HS), when the electron configuration of Fe³⁺ is t_{2g}⁵e_g⁰, t_{2g}⁴e_g¹ and t_{2g}³e_g², respectively (**Table 4**). That is, the spin electrons depend on the specific electron configuration rather than the valence state. Therefore, it is not suitable to establish the direct relationship between the spin states and the valence states.

To exclude the possible impact of conductivity raised by the reviewer, we measured electrical conductivities of SP-Fe₁-Ti and SD-Fe₁-Ti electrode at room temperature through the four probe DC technique. As shown in **Table R5**, no significant difference of the electrical conductivity in SP-Fe₁-Ti and SD-Fe₁-Ti was observed, which may be attributed to the excellent electrical conductivity of Ti substrate.

Table R5 was added into the revised Supporting Information as **Supplementary Table 1** and the related discussion was added in the revised manuscript (**Line 157-159, Page 6**) as follows:

“Moreover, the electrical conductivity of SP-Fe₁-Ti was very close to that of SD-Fe₁-Ti, benefiting from the excellent electrical conductivity of Ti substrate.”

Table R4. List of electron configuration and spin electrons for Fe³⁺ and Fe²⁺.

Valence state	Electron configuration	Spin electrons
Fe ³⁺	LS (t _{2g} ⁵ e _g ⁰)	1
	MS (t _{2g} ⁴ e _g ¹)	3
	HS (t _{2g} ³ e _g ²)	5
Fe ²⁺	LS (t _{2g} ⁶ e _g ⁰)	0
	MS (t _{2g} ⁵ e _g ¹)	2
	HS (t _{2g} ⁴ e _g ²)	4

Table R5. The electrical conductivity of electrode at room temperature.

Electrode	Electrical conductivity (kS cm ⁻¹)
SP-Fe ₁ -Ti	45

7. Why not provide the DEMS data of SD-Fe₁-Ti? If SP-Fe₁-Ti and SD-Fe₁-Ti have differences in the *NO reduction step, there should be a difference in their DEMS spectra.

Response: We thank the reviewer very much for this valuable suggestion. During the revision, we provided the DEMS data of SD-Fe₁-Ti in the revised Supporting Information. As shown in **Figure R22**, the peak intensity ratio (~3.92) of *NO to *NHO for SP-Fe₁-Ti was much smaller than that of SD-Fe₁-Ti (~5.83), suggesting an accelerated *NO hydrogenation process on the SP-Fe₁-Ti electrode.

During this revision, **Figure R22** was added into the revised Supporting Information as **Supplementary Figure 33** and related discussion was added in the revised manuscript (**Line 344-347, Page 13**) as follows:

*“Meanwhile, a much smaller peak intensity ratio (~3.92) of *NO to *NHO for SP-Fe₁-Ti than that of SD-Fe₁-Ti (~5.83, Supplementary Fig. 33) was observed, further confirming an accelerated *NO hydrogenation process on the SP-Fe₁-Ti electrode.”*

Figure R22. DEMS signals of NO and HNO during NITRR over SD-Fe₁-Ti electrode.

8. As shown in Figure 4e, the activation of Ti³⁺ caused by oxygen vacancies changes the bonding site from one to two, so how to define the number of oxygen vacancies in the DFT models? As known, it is easy to change the adsorption intensity of *NO or other intermediates by slightly changing the concentration of oxygen vacancies.

Besides, the authors stated that the higher binding energy of -1.8 eV for NO on SP-Fe1-Ti than that of -1.58 eV on SD-Fe1-Ti suggests that the Fe1-Ti pairs were more favorable for the deoxygenation of NO₃⁻ to *NO. Why could the adsorption energy of *NO act as a descriptor for NO₃⁻ reduction activity?

Response: Thanks a lot for the insightful comments. The number of oxygen vacancies (OVs) was determined based on the Gibbs free formation energy (G_{form}) of OVs. The G_{form} of OVs was defined as follows:

$$G_{\text{form}} = E_{\text{OV}2} - E_{\text{OV}1} + G_{\text{O}} \quad (1)$$

Where $E_{\text{OV}2}$ and $E_{\text{OV}1}$ were the total energies of the systems with (N+1) OVs and N OVs, respectively. G_{O} was the Gibbs free energy of an O atom with respect to H₂O and H₂, because the catalysts were synthesized under a reducing atmosphere (H₂/Ar) in the experiment. Equation 1 can be transformed into the following equation:

$$G_{\text{form}} = E_{\text{OV}2} - E_{\text{OV}1} + G_{\text{H}2\text{O}} - G_{\text{H}2} \quad (2)$$

The calculation of Gibbs free energy (G) for H₂O and H₂ was as follows:

$$G = E_{\text{DFT}} + E_{\text{ZPE}} - TS \quad (3)$$

Where E_{DFT} was the DFT calculated energy, E_{ZPE} was the zero-point energy calculated from the vibrational frequencies, T was set to the operating temperature for catalyst synthesis (673.15 K)

There were ten types of O atoms in amorphous TiO₂ surface in the optimized model (**Figure R23**). The calculated G_{form} of the first oxygen vacancy were shown in **Table R6**. According to the calculation results, the most stable configuration was the O10 vacancy with a G_{form} of -1.29 eV, indicating the highly possible formation of O10 vacancy. Therefore, the model was constructed with single Fe atom deposited at the O10 vacancy. Subsequently, the G_{form} of the second oxygen vacancy was based on the structure of Fe deposited at the O10 vacancy. The G_{form} of the second oxygen vacancy formed at the remaining sites was listed in **Table R7**. Obviously, the G_{form} of the second oxygen vacancy was significantly greater than 0 eV, implying that the second oxygen vacancy formation was thermodynamically unfavorable under the experimental conditions. Therefore, the model with single oxygen vacancy was used.

We chose the NO adsorption as the descriptor because NO is a key intermediate that determines the selectivity and activity of electrochemical denitrification. (ACS Catal. 2020, 10, 9320; ACS Catal. 2022, 12, 1394; Angew. Chem. Int. Ed. 2015, 54, 8255; ACS Catal. 2017, 7, 3869; Electrochim. Acta 2005, 50, 4318; J. Phys. Chem. Lett. 2021, 12, 6988; Nat. Commun. 2022, 13, 2338; Angew. Chem. Int. Ed. 2021, 60, 21966.) Recently, Greeley et al. investigated the effect of binding strength of NO adsorption on transition metals on product selectivity using DFT calculations, and found that the selectivity of complex NO_x electrochemical reduction could be described by the simple NO binding energy (ACS Catal. 2022, 12, 1394.). Meanwhile, the binding strength of NO was linearly correlated with that of N atom (Nat. Commun. 2022, 13, 2338, Fig. S2; Angew. Chem. Int. Ed. 2021, 60, 21966, Fig. S2&6), indicating there was also linear adsorption–energy scaling relationship between intermediates and NO. Accordingly, we conclude that the NO adsorption can serve as a descriptor for NITRR activity (Nat. Commun. 2022, 13, 2338; Angew. Chem. Int. Ed. 2021, 60, 21966.). Moreover, the stronger binding of *NO can provide a larger driving force for NO₃⁻ deoxygenation to *NO

Figure R23. Top view of oxygen vacancies on amorphous TiO₂ surface.

Table R6. the Gibbs free formation energy (G_{form}) of the first oxygen vacancy on amorphous TiO₂.

OVs	O1	O2	O3	O4	O5	O6	O7	O8	O9	O10
$G_{\text{form}}(\text{eV})$	-1.21	0.82	-1.25	-0.61	-0.46	0.37	0.02	0.61	-0.94	-1.29

Table R7. the Gibbs free formation energy (G_{form}) of the second oxygen vacancy on amorphous SP-Fe₁-Ti.

OVs	O1	O2	O3	O4	O5	O6	O7	O8	O9
$G_{\text{form}}(\text{eV})$	0.69	1.06	1.50	0.20	1.71	0.95	0.84	2.43	0.43

Reviewer #3:

This study presents the remarkable performance of a spin-promoted Fe₁-Ti pair on the oxygen vacancy-enriched TiO_x (SP-Fe₁-Ti) in the electrochemical nitrate reduction reaction (ENRR). In stark contrast to the spin-depressed Fe₁-Ti pair on the oxygen vacancy-rare TiO_x (SD-Fe₁-Ti), the SP-Fe₁-Ti catalyst demonstrated an impressive ammonia yield rate and nearly 100% selectivity for ammonia production across a wide range of potentials. Notably, high-quality NH₄Cl was successfully synthesized through continuous reaction. These findings highlight the significance of spin polarization in electrocatalysts for enhancing the performance of ENRR and provide valuable insights for the rational design of ENRR electrocatalysts. The manuscript exhibits a well-structured and well-written presentation of the research. Addressing the following concerns would further strengthen the paper for potential acceptance by Nature Communications.

1. My main concern is whether the discrepancies in the levels of oxygen vacancy and Ti³⁺ and the degree of crystallization also contribute to the differences in ENRR performance between SP-Fe₁-Ti and SD-Fe₁-Ti, in addition to the impact of spin polarization on the significantly improved performance of SP-Fe₁-Ti.

Response: We thank the reviewer very much for this constructive comment. First, we would like to re-emphasize that the spin-polarized Ti³⁺ sites were originated from the generation of oxygen vacancies (OVs), which has been fully confirmed by solid evidences of EPR, XPS and XAS in our original manuscript. For better understanding, we summarized the results as follows:

- (1) **EPR spectra.** OVs on both TiO_x/Ti and SP-Fe₁-Ti were accompanied by the appearance of spin-polarized Ti³⁺ species with an EPR signal at $g = 1.989$, which was invisible in the EPR spectra of SD-Fe₁-Ti without OVs (Fig. 1c). This difference indicated that the introduction of OVs could trigger the spin-polarization of Ti sites.
- (2) **XAS measurement.** The Ti L-edge soft-XAS that is highly sensitive to charge state, orbitals occupation and the spin orientation of electrons, was employed to reveal the formation of spin-polarized Ti³⁺ species on SP-Fe₁-Ti (Fig. 2d). In comparison with SD-Fe₁-Ti, the lower-energy shift and higher L₂ peak intensity of SP-Fe₁-Ti

verified the formation of spin-polarized Ti^{3+} species induced by OVs. The similar multiple spectral features of SD- Fe_1 -Ti with $SrTiO_3$ confirmed the Ti^{4+} valence state. (3) **XPS spectra.** Compared to SD- Fe_1 -Ti and Ti foam, the Ti 2p XPS spectra of SP- Fe_1 -Ti shifted to lower binding energy (Fig. 2e), further confirming the spin-polarized Ti^{3+} species on SP- Fe_1 -Ti.

Therefore, the spin-polarized Ti^{3+} species induced by OVs contributed to the improved NITRR activity of SP- Fe_1 -Ti electrode, which was emphasized in our manuscript. Compared with SD- Fe_1 -Ti, the unpaired spin electron of Ti atom on SP- Fe_1 -Ti was found to inject into *NHO and enhance the electron transfer from Fe-Ti spin pair to further stabilize *NHO, resulting in the remarkably decreased free energy of *NO to *NHO from 0.98 eV to -0.23 eV. To further highlight the impact of spin polarization, we calculated the free energy of *NHO intermediate adsorbed on the Fe_1 -Ti pairs involving a Ti atom without spin-polarization for comparison (Supplementary Fig. 32), and found that the calculated value of 0.37 eV was much higher than that on SP- Fe_1 -Ti, suggesting that the spin-polarization of Ti atom significantly contributed to the higher NITRR activity.

As for the contribution of crystallization degree to the NITRR performance, we conducted some control experiments in the original manuscript, and did not find any obvious difference in the surface amorphous titanium oxide layer (TiO_x) between SP- Fe_1 -Ti and SD- Fe_1 -Ti treated in the same calcination temperature (400 °C) as evidenced by high-resolution transmission electron microscopy (HRTEM) images (Supplementary Figs. 4 and 5), thus excluding the impact of crystallization degree towards NITRR performance of SP- Fe_1 -Ti and SD- Fe_1 -Ti.

2. The determination of the Fe dopant quantity in both SP- Fe_1 -Ti and SD- Fe_1 -Ti should be conducted using inductively coupled plasma (ICP) analysis. Additionally, it is recommended to compare the catalytic performance for ammonia yield based on mass activity, in addition to the electrochemical surface area (ECSA)-normalized activity. This comparison would provide a more comprehensive assessment of the catalytic performance of the materials.

Response: Thanks a lot for these constructive suggestions. The Fe quantity on the

SP-Fe₁-Ti and SD-Fe₁-Ti using inductively coupled plasma (ICP) analysis was determined to be about 0.06 wt.%. Actually, the ammonia yield rate normalized to the mass of Fe (the dominantly active center) on SP-Fe₁-Ti and SD-Fe₁-Ti was provided in the original manuscript (**Line 94-99, Page 4**) as follows.

“The SP-Fe₁-Ti electrode exhibits an excellent NITRR performance with an unprecedented NH₃ yield rate of 272000 μg h⁻¹ mg_{cat}⁻¹ at -0.4 V vs. RHE, far superior to the counterpart with spin-depressed Fe₁-Ti pairs (51000 μg h⁻¹ mg_{cat}⁻¹) and one order of magnitude higher than mostly reported NITRR electrocatalysts.”

3. Could you please provide information about the specific electrolyte used for NH₄Cl synthesis in the recovery chamber?

Response: We thank the reviewer for this valuable suggestion. During NH₄Cl synthesis process, 1 M HCl solution was used to recover the produced NH₃ from NO₃⁻ reduction. During the revision, the information was added to the revised manuscript (**Line 408-413, Page 15**) as follows:

“During operation, the NO₃⁻-containing solution (250 mL) as synthetic effluent was treated through electrochemical reduction on SP-Fe₁-Ti cathode, and circulated by a peristaltic pump with a flow rate of 60 mL min⁻¹. Then the NH₃-containing effluent was directly guided into the hollow polypropylene (PP) fiber arrays immersed into 1 M HCl solution (800 mL) to recover NH₃.”

4. On page 12, it is mentioned that the recovery of ammonium via ENRR on SP-Fe₁-Ti was tested at a current density of 200 mA cm⁻² for 200 h of electrolysis. There is a concern regarding whether the high current density and strong basic media could potentially compromise the hydrophobicity of the gas-permeable membrane. It is requested to provide the contact angle of the post-reacted membrane as a measure of its hydrophobicity.

Response: We thank the reviewer very much for these valuable suggestions. During the revision, we provided the contact angle of the pristine and post-reacted membrane in the revised manuscript. After long-term electrolysis, the measured contact angle of membrane was determined to be 124.7 ° (**Figure R24**), very close to

that of pristine membrane (123.5°). These results suggested that the reacted membrane could kept its hydrophobic nature.

During this revision, **Figure R24** was added into the revised Supporting Information as **Supplementary Figure 38** and the related discussion was added in the revised manuscript (**Line 421-424, Page 15 and 16**) as follows:

“After long-term electrolysis, the measured contact angle of membrane (Supplementary Fig. 38) was determined to be 123.5°, similar with that of pristine membrane (123.5°), suggesting that the post-reacted membrane still kept its hydrophobicity.”

Figure R24. The contact angle of (a) the pristine and (b) post-reacted membrane.

5. In Figure 1c and Figure 2e, there are observed to be slight differences in the electron paramagnetic resonance (EPR) spectra and high-resolution Ti 2p X-ray photoelectron spectroscopy (XPS) spectra between TiO_x/Ti and SP-Fe₁-Ti. More explanations are suggested.

Response: We thank the reviewer very much for this valuable suggestion. The slight differences in the electron paramagnetic resonance (EPR) spectra between TiO_x/Ti and SP-Fe₁-Ti may be arisen from the Fe loading, which occupied the oxygen vacancies (OVs) along with the slight decrease of OVs intensity. The tiny differences in the high-resolution Ti 2p X-ray photoelectron spectroscopy (XPS) spectra between TiO_x/Ti and SP-Fe₁-Ti could be ascribed to the very weak interaction between Fe and Ti, suggesting that the Fe loading did not significantly change the electronic state of Ti on the TiO_x/Ti.

6. Figure 2e and Figure 2f exhibit a partial overlap in their data. It is recommended to revise the positioning of the data in these figures to ensure better visibility and clarity of the information presented.

Response: We thank the reviewer for this kindly reminder. During the revision, we carefully revised the position of the data in Figure 2e and Figure 2f to ensure the data visibility and the information clarity.

7. In Figure 3b, the observed Faradaic efficiency (FE) being less than 100% suggests the occurrence of side reactions that contribute to the incomplete conversion of nitrate. Two potential side reactions that could affect the FE are the hydrogen evolution reaction (HER) and the nitrate-to-nitrite reaction. Please identify their contributions.

Response: We thank the reviewer very much for this valuable suggestion. During the revision, the contributions of hydrogen evolution reaction (HER) and the nitrate-to-nitrite reaction on the electrode to the faradaic current were further evaluated. We investigated possible gas and liquid products using gas chromatography (GC, Thermal Trace-1300) equipped with thermal conductivity detector (TCD) and UV-Vis spectrometry. As shown in **Figure R25**, the only detected gas product was H₂, and the only detected liquid product was NO₂⁻. The Griess test was used to determine the NO₂⁻ concentration in electrolytes after the potentiostatic test. Generally, N-1-naphthyl ethylenediamine dihydrochloride (0.04 g), p-aminobenzene sulfonamide (0.8 g), and H₃PO₄ (2 mL, 85%) were dissolved in DI water (10 mL) as the Griess reagent. The diluted electrolyte (5 mL) was mixed with the Griess agent (100 μL) and rested for 10 min at room temperature to conduct UV-vis tests. NO₂⁻ concentrations of the electrolyte were determined by the absorbance at 540 nm. In the same operation, various NaNO₂ aqueous solutions were used as the standard samples to obtain the calibration curve (**Figure R26**).

The FE of product *i* (FE_{*i*}) was calculated via the following relation:

$$FE_i = (n_i \times F \times c_i \times V) / (17 \times Q)$$

where *n_i* is the number of electrons transferred to product *i*, *c_i* is the measured product *i* concentration, *V* is the volume of the electrolyte or the upper space, *F* is the Faraday constant (96485 C mol⁻¹) and *Q* is the total charge.

As shown in **Figure R27**, the FEs of H_2 and NO_2^- were calculated to be 0.7% and 4.1%, respectively. After taking the two side-reactions into consideration, the total FE was 100% as expected.

During this revision, **Figure R25-27** were added into the revised Supporting Information as **Supplementary Figure 17-19** and related discussion was added in the revised manuscript (**Line 243-248, Page 9 and 10**) as follows:

“The contributions of other possible products to the faradaic current were also evaluated using gas chromatography equipped with thermal conductivity detector (TCD) and UV-Vis spectrometry. Only H_2 and NO_2^- were detected (Supplementary Fig. 17 and Supplementary Fig. 18), and their FEs were calculated to be 0.7% and 4.1%, respectively, at -0.4 V vs. RHE. After taking the two side-reactions into consideration, the total FE (Supplementary Fig. 19) was 100% as expected.”

Figure R25. Representative GC of gas products obtained on SP-Fe₁-Ti electrode at -0.4 V vs. RHE.

Figure R26. Standard calibration curves for UV-Vis detection of NO_2^- from the Griess's method (a) the standard solutions (b) raw UV-Vis spectra (c) linear calibration.

Figure R27. The FE of H₂ and NO₂⁻ and NH₃ during NITRR on on SP-Fe₁-Ti electrode at -0.4 V vs. RHE.

REVIEWER COMMENTS

Reviewer #1 (Remarks to the Author):

The authors have diligently addressed the critical comments from the first round of the review and have substantially improved the quality of their manuscript. The catalyst-design strategy presented is indeed highly promising, and the reviewer is happy to recommend the publication of this quality paper in Nature Communication.

Reviewer #2 (Remarks to the Author):

In the previous major revision, the authors addressed the reviewer's concerns about the spin-polarized effects of Fe-Ti pairs for the enhanced NITRR. However, the reviewer thinks some of the answers are insufficient. Therefore, the reviewer will not suggest accepting this manuscript unless the following comments are carefully fulfilled.

1. When the NH₃ yield rate is first mentioned in the abstract, introduction, and results, the authors should note that the NH₃ yield rate is normalized to the mass of Fe atoms on the Ti substrate. If not, the readers will be misled. In the abstract, the authors should write the accurate value of NH₃ FE (95.9%) rather than "almost 100%". Besides, the description of "one order of magnitude higher than mostly reported NITRR electrocatalysts" is not reasonable because most of the reported electrocatalysts did not use a tiny component of the catalysts to define the NH₃ yield rate.

2. The authors gave several possible reasons for the observed 200 mA cm⁻² at -0.4 V in the flow-through electrolyzer, such as using commercial dimension stable anode (DSA), the large electrode size, the accelerated mass transfer, and reduced ohmic losses.

What is the composition of DSA? It is not reasonable to denote the active anode as DSA.

It is still surprising that the current density of the flow-through electrolyzer can increase almost three times compared to the H-type cell due to the use of DSA and accelerated mass transfer. Generally, the cathodic current mainly depends on the applied potential versus the reference electrode. The anode performance only impacts the total voltage of the electrochemical cell. If the anode shows extremely poor performance, the voltage of the whole cell would be larger than the voltage range of the potentiostats, which makes it impossible to implement cathodic electrolysis. So, the authors should

show a figure and the structure of the flow-through electrolyzer to explain the three times higher current density at -0.4 V.

3. According to the reported DFT calculations, the NO_3^- is adsorbed on the catalytic sites exclusively by its oxygen atoms, and the NO is adsorbed on the sites by its nitrogen atom for further NH_3 generation. Therefore, there is no scaling relationship between the binding strengths of NO_3^- and $^*\text{NO}$ on the catalytic sites, respectively. Moreover, $^*\text{NO}_2$ is the intermediate between NO_3^- and $^*\text{NO}$, so the NO_3^- deoxygenation to $^*\text{NO}$ cannot bypass the $^*\text{NO}_2$. Why can the stronger binding of $^*\text{NO}$ directly provide a more significant driving force for NO_3^- deoxygenation to $^*\text{NO}$?

4. There is a mistake in Figure 2b. The Fe $2p_{3/2}$ binding energy of SP-Fe1-Ti should be ~ 723.4 eV, not the marked 720.4 eV.

Reviewer #3 (Remarks to the Author):

The authors have well addressed my concerns and accordingly I recommend its acceptance for publication in Nature Communications.

Point-by-point responses to the reviewers' comments

First of all, we thank the reviewers for their valuable comments and suggestions, which are very helpful to improve the quality and clarity of this paper. To address the specific concern/point clearly, we have separated the referees' comments into question areas and answered them in turn as follows.

Reviewer #1:

The authors have diligently addressed the critical comments from the first round of the review and have substantially improved the quality of their manuscript. The catalyst-design strategy presented is indeed highly promising, and the reviewer is happy to recommend the publication of this quality paper in Nature Communication.

Response: We thank the reviewer for this constructive review process and strong support on the publication of this work.

Reviewer #2:

In the previous major revision, the authors addressed the reviewer's concerns about the spin-polarized effects of Fe-Ti pairs for the enhanced NITRR. However, the reviewer thinks some of the answers are insufficient. Therefore, the reviewer will not suggest accepting this manuscript unless the following comments are carefully fulfilled.

1. When the NH_3 yield rate is first mentioned in the abstract, introduction, and results, the authors should note that the NH_3 yield rate is normalized to the mass of Fe atoms on the Ti substrate. If not, the readers will be misled. In the abstract, the authors should write the accurate value of NH_3 FE (95.9%) rather than "almost 100%". Besides, the description of "one order of magnitude higher than mostly reported NITRR electrocatalysts" is not reasonable because most of the reported electrocatalysts did not use a tiny component of the catalysts to define the NH_3 yield rate.

Response: We thank the reviewer very much for the valuable comments and suggestion. During the revision, we underlined that the NH_3 yield rate was normalized to the mass of Fe atoms on the Ti substrate when the NH_3 yield rate is first mentioned

in the abstract (**Line 31 and 32, Page 2**), introduction (**Line 98 and 100, Page 4**), results (**Line 222, Page 9**) and discussion (**Line 442, Page 16**). To avoid misunderstanding, we revised the statement of “almost 100% NH₃ Faradic efficiency” to “a high NH₃ Faradic efficiency of 95.2 %” (**Line 31, Page 2; Line 97, Page 4; Line 222, Page 9, and Line 442, Page 16**) and the “one order of magnitude higher than mostly reported NITRR electrocatalysts” to “the mostly reported NITRR electrocatalysts” (**Line 32, Page 2; Line 100, Page 4**) in the revised manuscript.

2. The authors gave several possible reasons for the observed 200 mA cm⁻² at -0.4 V in the flow-through electrolyzer, such as using commercial dimension stable anode (DSA), the large electrode size, the accelerated mass transfer, and reduced ohmic losses. What is the composition of DSA? It is not reasonable to denote the active anode as DSA. It is still surprising that the current density of the flow-through electrolyzer can increase almost three times compared to the H-type cell due to the use of DSA and accelerated mass transfer. Generally, the cathodic current mainly depends on the applied potential versus the reference electrode. The anode performance only impacts the total voltage of the electrochemical cell. If the anode shows extremely poor performance, the voltage of the whole cell would be larger than the voltage range of the potentiostats, which makes it impossible to implement cathodic electrolysis. So, the authors should show a figure and the structure of the flow-through electrolyzer to explain the three times higher current density at -0.4 V.

Response: Thanks a lot for these valuable comments and suggestions. We are sorry for unclear statement regarding the composition of DSA. In this study, the DSA used in the flow-through electrolyzer was a mesh-type Ru_{0.8}Ir_{0.2}O₂/Ti electrode with a coating thickness of ~10 μm. For better understanding, more detailed information regarding the composition of the DSA used in the flow-through electrolyzer was added in the revised manuscript (**Line 414 and 415, Page 15**). Moreover, as suggested by reviewer, we offer a picture and the corresponding schematic diagram to illustrate the structure of the flow-through electrolyzer. As displayed in **Figure R1**, the electrolyzer was an undivided cell of cathode and anode chambers (internal dimensions: 7 × 7 × 5 cm³) made of plexiglass (poly (methyl methacrylate), PMMA). The SP-Fe₁-Ti cathode (2 × 2

cm²) and the DSA anode (2 × 2 cm²) were fixed into the chambers in a more compact manner (2-cm spacing) and separated by the nafion film (4.5 × 4.5 cm²). The hollow polypropylene (PP) fibers were assembled into the home-made fiber arrays, which were placed in acidic solution to act as an NH₃ recovery reactor. The two open ends of the membrane arrays were fixed with epoxy resin and were connected with the electrolyzer by rubber tube. During operation, nitrate-containing wastewater flowed through the porous SP-Fe₁-Ti cathode to be reduced into NH₃-containing wastewater, and then directly guided into the hollow PP fiber arrays immersed into 1 M HCl solution to recover NH₃. When the solution penetrated directly through the electrode pores with a flow-through mode, the diffusional boundary layer thickness could be reduced, thus promoting mass transfer rate and reaction current (Acc. Chem. Res. 2019, 52, 596; Environ. Sci. Technol. 2021, 55, 12596). Moreover, a compact structure of electrolytic reactor was able to reduce ohmic losses and subsequently increase the current density of device (Joule 2021, 5, 1776; Water Res. 2023, 242, 120256). Therefore, the flow-through mode and the compact cell structure could significantly accelerate the mass transfer and reduce ohmic losses of reactions, collectively resulting in a higher current density compared to that of in a H-type cell.

For better understanding, during this revision, **Figure R1** was added into the revised Supporting Information as **Supplementary Figure 38** and related discussion was added in the revised Supporting Information (**Line 299–307, Page 41**) as follows:

“The electrolyzer was an undivided cell of cathode and anode chambers (internal dimensions: 7 × 7 × 5 cm³) made of plexiglass (poly(methyl methacrylate), PMMA). The SP-Fe₁-Ti cathode (2 × 2 cm²) and the DSA anode (2 × 2 cm²) were fixed into the chambers in a more compact manner (2-cm spacing) and separated by the nafion film (4.5 × 4.5 cm²). The hollow polypropylene fibers were assembled into the home-made fiber arrays, which were placed in acidic solution to act as an NH₃ recovery reactor. The two open ends of the membrane arrays were fixed with epoxy resin and were connected with the electrolyzer by rubber tube.”

Figure R1. (a) A schematic diagram of the integrated device composed of a flow-through NITRR electrolyzer and a membrane-based ammonia recovery unit for simultaneous nitrate electroreduction and ammonia recovery. (b) Photograph of the integrated device for simultaneous nitrate electroreduction and ammonia recovery.

3. According to the reported DFT calculations, the NO_3^- is adsorbed on the catalytic sites exclusively by its oxygen atoms, and the NO is adsorbed on the sites by its nitrogen atom for further NH_3 generation. Therefore, there is no scaling relationship between the binding strengths of NO_3^- and $^*\text{NO}$ on the catalytic sites, respectively. Moreover, $^*\text{NO}_2$ is the intermediate between NO_3^- and $^*\text{NO}$, so the NO_3^- -deoxygenation to $^*\text{NO}$ cannot bypass the $^*\text{NO}_2$. Why can the stronger binding of $^*\text{NO}$

directly provide a more significant driving force for NO₃⁻ deoxygenation to *NO?

Response: We thank the reviewer very much for the valuable comments and suggestion. Actually, NO₃⁻ to *NO underwent an NO₃⁻ adsorption step and two deoxygenation steps, namely, *+NO₃⁻ → *NO₃ → *NO₂ → *NO (**Figure R2**). In the case of the SD-Fe₁-Ti system, the NO₃⁻ adsorption underwent an uphill Gibbs free energy change of 0.34 eV. Afterward, *NO₃ first deoxygenated to *NO₂ with a downhill free energy change of -1.52 eV. Subsequently, *NO₂ continued to remove one oxygen to form *NO with an energy release of 1.81 eV. As for the SP-Fe₁-Ti system, the above three steps release the energy of 0.64, 1.65, and 0.93 eV, respectively. Finally, NO₃⁻ deoxygenated to *NO with a total downhill Gibbs free energy change of -3.00 eV on SD-Fe₁-Ti, while the total downhill free energy change for the above steps was -3.22 eV on SP-Fe₁-Ti. Therefore, NO₃⁻ deoxygenation to *NO on SP-Fe₁-Ti was more favorable thermodynamically.

For better understanding, during this revision, related discussion was added in the revised manuscript (**Line 336–347, Page 13**) as follows:

*“Actually, NO₃⁻ to *NO underwent an NO₃⁻ adsorption step and two deoxygenation steps, namely, *+NO₃⁻ → *NO₃ → *NO₂ → *NO. In the case of the SD-Fe₁-Ti system, the NO₃⁻ adsorption underwent an uphill Gibbs free energy change of 0.34 eV. Afterward, *NO₃ first deoxygenated to *NO₂ with a downhill free energy change of -1.52 eV. Subsequently, *NO₂ continued to remove one oxygen to form *NO with an energy release of 1.81 eV. As for the SP-Fe₁-Ti system, the above three steps release the energy of 0.64, 1.65, and 0.93 eV, respectively. Finally, NO₃⁻ deoxygenated to *NO with a total downhill Gibbs free energy change of -3.00 eV on SD-Fe₁-Ti, while the total downhill free energy change for the above steps was -3.22 eV on SP-Fe₁-Ti. Therefore, NO₃⁻ deoxygenation to *NO on SP-Fe₁-Ti was more favorable thermodynamically.”*

To avoid misunderstanding, we deleted the statement of *“These results suggested that the Fe₁-Ti pairs in spin polarization were more favorable for the deoxygenation of NO₃⁻ to *NO.”* and revised the statement of *“Conclusively, both spin-polarized Fe-Ti pair sites can effectively and stabilize *NO and *NHO intermediates to facilitate the deoxygenation of NO₃⁻ and the subsequent *NO hydrogenation”* to *“Conclusively,*

both spin-polarized Fe-Ti pair sites can effectively facilitate the deoxygenation of NO_3^- to $^*\text{NO}$ and stabilize $^*\text{NO}$ and $^*\text{NHO}$ intermediates to boost the subsequent $^*\text{NO}$ hydrogenation" (Line 402-404, Page 15) in revised manuscript.

Figure R2. Calculated free energy diagrams for the steps of $^*\text{+NO}_3^- \rightarrow ^*\text{NO}$ on SD-Fe₁-Ti and SP-Fe₁-Ti.

4. There is a mistake in Figure 2b. The Fe 2p_{3/2} binding energy of SP-Fe₁-Ti should be ~723.4 eV, not the marked 720.4 eV.

Response: We are very sorry for this typo error. During the revision, we revised the marked Fe 2p_{3/2} binding energy of SP-Fe₁-Ti in Figure 2b to 723.4 eV.

Reviewer #3:

The authors have well addressed my concerns and accordingly I recommend its acceptance for publication in Nature Communications.

Response: We thank the reviewer for this constructive review process and strong support on the publication of this work.

REVIEWERS' COMMENTS

Reviewer #2 (Remarks to the Author):

The authors have addressed my concerns, and I recommend accepting this manuscript.